# Timing along the cardiac cycle modulates neural signals of reward-based learning

Elsa F. Fouragnan [1,2,3] ✉, Billy Hosking[2,3], Yin Cheung[1], Brooke Prakash[1], Matthew Rushworth [1,6] & Alejandra Sel[1,4,5,6]

Natural fluctuations in cardiac activity modulate brain activity associated with sensory stimuli, as well as perceptual decisions about low magnitude, near-threshold stimuli. However, little is known about the relationship between fluctuations in heart activity and other internal representations. Here we investigate whether the cardiac cycle relates to learning-related internal representations – absolute and signed prediction errors. We combined machine learning techniques with electroencephalography with both simple, direct indices of task performance and computational model-derived indices of learning. Our results demonstrate that just as people are more sensitive to low magnitude, near-threshold sensory stimuli in certain cardiac phases, so are they more sensitive to low magnitude absolute prediction errors in the same cycles. However, this occurs even when the low magnitude prediction errors are associated with clearly suprathreshold sensory events. In addition, participants exhibiting stronger differences in their prediction error representations between cardiac cycles exhibited higher learning rates and greater task accuracy.

In situations where we must make decisions based on noisy or incomplete information - for example deciding whether to cross the street on a foggy morning with poor visibility - our choices can be modulated, albeit to a small degree, by the timing of the cardiac cycle. Studies investigating near-threshold sensory events, like visual, auditory or somatosensory events, have shown that timing in the cardiac cycle (e.g., whether events happen during the systolic or diastolic phases of the cardiac cycle) impacts the perception and reaction to sensory cues through changes in associated neural signals[1–3]. Although heart-brain interactions are starting to be understood in relation to sensory-driven processes, it is unclear whether the cardiac cycle has a similar relationship with other internal representations which are non-sensory but which, like sensory stimuli, mediate decision-making[4–9]. Here we focus on a much-studied internal representation – the reward prediction error [PE][10] – and investigate whether the cardiac cycle also

determines the impact that each PE will have on learning. Importantly, the magnitude of the PE can be dissociated from the magnitude of the accompanying sensory stimulus. This makes it possible to determine whether the cardiac cycle has an impact on near-threshold PEs even if the PEs are associated with clearly suprathreshold sensory stimuli.

Adaptive decisions rely on accurate subjective value estimates associated with past experience of choices and their consequences. These values can be formally defined through the reinforcement learning framework[11] that uses the difference between expectation and outcome (the PE) to update values associated with choices. A choice that led to a positive outcome is more likely to be associated with a higher value than a choice that did not. While the signed PE represents how much better or worse the value of an outcome is compared to what was expected, the absolute PE (also called 'salience', 'surprise', or 'unsigned PE') represents how much an outcome differs from

[1]Wellcome Centre for Integrative Neuroimaging (WIN), Department of Experimental Psychology, University of Oxford, Oxford OX1 3UD, UK. [2]Brain Research Imaging Centre (BRIC), Faculty of Health, University of Plymouth, Plymouth PL6 8BU, UK. [3]School of Psychology, Faculty of Health, University of Plymouth, Plymouth PL4 8AA, UK. [4]Centre for Brain Science, Department of Psychology, University of Essex, Wivenhoe Park, Colchester CO4 3SQ, UK. [5]Essex ESNEFT Psychological Research Unit for Behaviour, Health and Wellbeing, University of Essex, Wivenhoe Park, Colchester CO4 3SQ, UK. [6]These authors contributed equally: Matthew Rushworth, Alejandra Sel. ✉e-mail: elsa.fouragnan@plymouth.ac.uk

expectations regardless of whether it is better or worse[12]. Activity in separate neural networks has been related to the signed PE and absolute PE[13,14]. It has thus been hypothesised that these two quantities represent two different dimensions of learning. Whereas positive and negative signed PEs lead to the reinforcement or extinction of the choices that led to them[15], the absolute PEs can determine the extent to which the associations between outcome and expectations need to be adjusted[13,16]. Even if a choice leads to a clearly suprathreshold sensory event, the PE it entails might be large, small, or even near-threshold depending on what the decision maker's prior expectations were. This means that we can examine whether near-threshold PEs relate to the cardiac cycle even if they are associated with suprathreshold sensory events.

The cardiac cycle is a series of contractions and relaxations that help the heart pump blood throughout the body. Each cardiac cycle has a diastolic phase (also known as diastole) in which the heart chambers relax and fill with blood, and a systolic phase (also known as systole) in which the heart chambers contract and pump blood to the periphery. These two physiological phases are differentially signalled to the brain through baroreceptor firing during systole and by a pause in firing during diastole. These signals are linked to activity in brainstem regions such as periaqueductal grey as well as forebrain regions such anterior cingulate cortex (ACC), anterior insula (AI), amygdala, and orbitofrontal cortex (OFC)[17,18]. Behavioural and neuroimaging research suggests that sensory perception and executive control are affected differently by the heart phase[2,19,20]. Although such a distinction remains debated[21], the studies show that participants are more sensitive to perceiving visual, auditory and somatosensory signals during diastole and less sensitive during systole when key sensory brain regions receive cardiac-related afferent signals increasing the excitability levels in these regions. By contrast, executive processes such as attention switching, active sampling and motor control may be enhanced during systole as opposed to diastole. This suggests that different cognitive processes may be prioritised at different points in the cardiac cycle[3,22].

Learning is affected by states of cognitive and physiological arousal that can fluctuate over time[23]. For example, it is long established that heart rate slows down in situations such as learning that require attention to the environment[24]. Although the exact functional role of cardiac deceleration on cognition is still under debate[25,26], heart rate deceleration - which involves longer diastolic phase, is appreciated as a physiological mechanism that better prepares the organism to take in sensory stimuli and respond to them[27]. Our aim in the current study is to examine the relationship between the cardiac cycle and quantitative indices of the learning process such as signed and absolute PEs. Model estimates of signed and absolute PE can capture cognitive and physiological fluctuations as learning progresses. Although model estimates are good predictors of behavioural change[28], studies exploiting concurrent trial-by-trial physiological changes can offer additional explanatory power when analysing behaviours or neural data related to signed and absolute PE. For example, some studies have used changes in eye gaze or pupil dilation to disentangle attentional and learning processes involved in PE coding[29]. Others have used single-trial variability in EEG to expose latent brain states related to PE, thereby complementing more conventional model-based fMRI analyses. Using such trial-by-trial estimates has revealed the temporal and spatial neural correlates of these learning signals in human and animal brains[13,30]. Signed PE-related activity has been reported in a number of brain areas but absolute PE-related activity has been most often linked to ACC and AI[13,29,31–33] – in or adjacent to brain areas associated with cardiac-related activity[22]. In addition, recent studies have shown that absolute PE-related activity appears shortly after the outcome while signed PE-related activity has a longer latency after the outcome[13,30].

Because absolute PE information is neurally encoded early after outcome onset and given the adjacency of brain areas encoding both

saliency and cardiac activity, we hypothesised that cardiac signals might interact with the impact that the absolute PE, as opposed to the signed PE, has on learning. By analogy with the variation in the impact of near-threshold perception that occurs in relation to the cardiac cycle, we investigated variation in the impact of near-threshold absolute PEs on learning as a function of the cardiac cycle. We did this by capitalising on a whole-brain machine learning technique and high temporal resolution data; we exploit trial-by-trial variability in the cardiac-related signal to investigate separately how changes in absolute PE and signed PE throughout the task are modulated (enhanced or decreased) by the two cardiac phases. We hypothesise that the timing within the cardiac cycle (e.g., whether a decision outcome occurs at cardiac diastole or cardiac systole) modulates the strength of the neural representation of the outcome. In line with previous evidence showing a better ability to perceive information during diastole as opposed to systole[2,21], we hypothesise that near-threshold absolute PE events regardless of their perceptual magnitude will be better represented at diastole than systole. If this is the case, then internal subjective representations of choice value will reflect time points within the cardiac cycle when they are constructed.

Here, we first show that there is an intrinsic relationship between the absolute PE dimension of decision outcome and the HEP, which we refer to as absPE-HEP. In addition, we demonstrate that the timing of a reward-related outcome, with respect to the cardiac cycle, mediates the absPE-HEP magnitude as well as learning and overall performance in the task. More specifically, absPE-HEP during the first HEP after outcome, was lower if the outcome happened at systole than diastole. Across participants, this difference was related to learning rates in the computational model and ultimately performance as indexed by a simple, computational model-independent measure – the number of rewards received in the task. Furthermore, the relationship between absPE-HEP and learning is only observed in the block of the task where learning was possible.

## Results
### Statistics of the reward environment predict learning
Participants carried out a reward-guided decision task inspired by previous credit assignment tasks[34,35]. On each trial, subjects were shown two visual cues that are associated with different category-specific brain areas (a face and a house) and asked to predict which colour (orange or blue) was most likely to follow (Fig. 1a, b). Participants made choices by pressing corresponding left or right buttons to indicate their prediction of orange or blue. The actual outcome (a single colour) was then displayed. Participants were instructed that the chance of the correct colour being blue or orange depended only on the cue-outcome prediction strength and the recent outcome history. While participants performed the task, we recorded neural responses to heartbeats with EEG and ECG (Fig. 1a) during the outcome which included a four-second period to ensure that multiple heartbeats would be recorded (mean: 4.58, std ±0.85). To modulate learning throughout the task, participants performed tasks employing four association schemes presented in separate blocks. There were three predictive schemes with high associations between cues and colours (highly predictive anticorrelated [HCA], highly predictive correlated [HPC], and variable predictive schemes [VP]) and one scheme with no associations between cues and colours (non-predictive scheme [NP]) (Fig. 1c). Initial analyses confirmed that the cue-colour prediction strength (which determined reward contingencies) was the primary modulator of adaptive behaviour in the task including performance and reaction time (Fig. 1f, g). Indeed, and as expected, participants reaction times were faster in HPA blocks than the others and generally more accurate in the highly predictive blocks (HPA and HPC) compared to the other two (VP and NP).

To investigate the two dimensions of learning (signed and absolute PE), we modelled participants' choices using four reinforcement

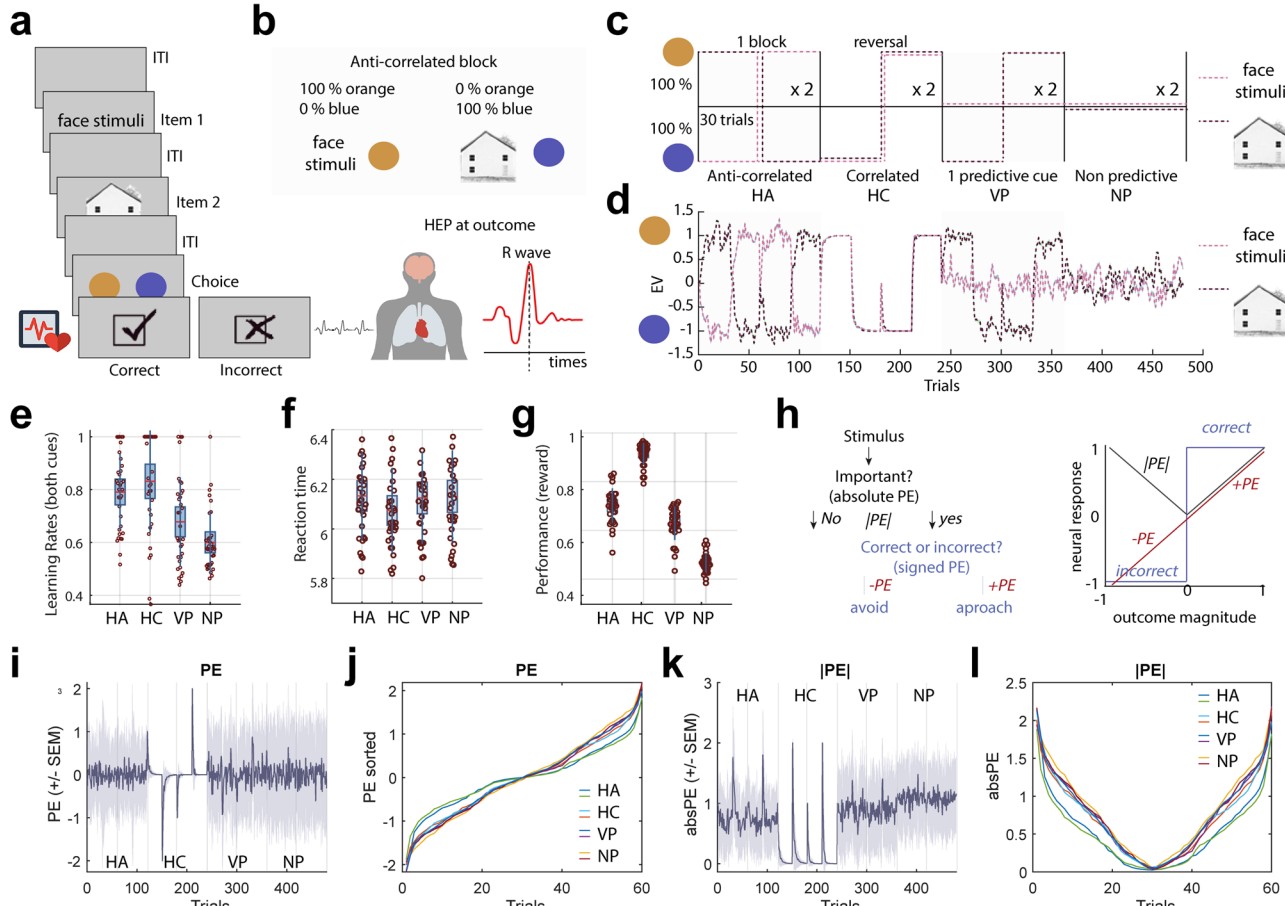

Fig. 1 | Schematic representation of the task and RL results for all four association schemes, highly predictive anticorrelated (HA), highly predictive correlated (HC), variable predictive (VP) and non-predictive (NP). a Representation of the task and cardiac-related neural signals (HEP) recorded at outcome for 4 s - enough to record on average 3.5 HEPs. b Example of association between cues and predicted colour for one version of the task (anti-correlated blocks – see "Methods" section). c Prediction strengths for each block. d Model prediction of choices. e Learning rates from the best fitting model. f, g Reaction times and performance (mean reward across the four blocks) h left panel Definition of absolute and signed PE. h Right panel Each coloured pattern illustrates a different way in which activity

in a neural structure may manifest if it is sensitive to different aspects of outcomes and their associated PE (locked at time of feedback). The blue pattern illustrates activity as a function of outcome valence – in a categorical manner as either positive or negative. The red pattern shows a monotonically increasing response profile consistent with a continually varying PE representation. The black pattern is continually varying as a function of the difference between the outcome and the expectation in an unsigned fashion. i Signed PEs are illustrated after averaging across all participants and all blocks, for the example of a task like that illustrated in panel (c). j Absolute PEs are similarly illustrated after averaging across participants and across blocks (k, l) and using the same format as for panels (i, j).

learning models which differed in few ways. The first model learns simple cue values for face and house (Simple Cue model). The second model has a recency weighting at the time of learning which updated the value estimate for the second of the two presented cues more than the first. The third model learns expected value for the differential pair of cues and the last one has trial-wise learning rates, which we refer to as the dynamic learning rate model (see "Computational modelling" section). For all models, we estimated free parameters by likelihood maximisation and Laplace approximation of model evidence to calculate the integrated Bayesian Information Criterion (BIC) and the exceedance probability respectively (this can now be found in Supplementary Fig. 1A–C). Bayesian model comparison revealed that a Simple Cue model explains the data better (Lower BIC values indicate better model fit, SimC model: BIC = 9350, RW model: BIC = 9370, DYNA model: BIC = 9400, CUNJ: BIC = 9390). Exceedance probabilities for the models based on approximate posterior probabilities suggested that our Simple Cue model outperformed the others (φ = 0.95; Supplementary Fig. 1C insert). Having established the goodness of fit of the Simple Cue model to behaviour (Supplementary Fig. 1D) all further analyses were conducted using the outcome-related signals estimated with this model (Fig. 1d, e, h, l). Parameter recovery[36,37] was

also performed on the best model and presented in Supplementary Fig. 1E and Supplementary Table 1.

## Grand average modulation of cardiac-related neural signals in learning-related dimensions

We next looked for EEG signatures of the heartbeat-evoked potential (HEP). Figure 2a presents the topographical characteristics of the HEP based on the averaged HEP recorded during the outcome (see "Methods" section for the construction of the ERP for the HEP). A morphology analysis revealed that the HEP was widely distributed along frontocentral and centro-parietal areas including the following spatial regions (Frontocentral sites: F1, Fz, F2, FC1, FCz, CF2; Centro-parietal sites: C1, Cz, C2, CP1, CPz, CP2 as in Fig. 2a). We then probed whether these EEG HEP signatures were modulated as a function of learning. To do this, we looked at two dimensions of learning as well as outcome valence. The two dimensions of learning included the fully parametric signed PE signal and the absolute PE also called salience (which connotes how surprising an outcome is). This approach has the advantage of looking at two orthogonal RL signals as seen in Fig. 1h. The outcome valence was simply correct versus incorrect outcomes.

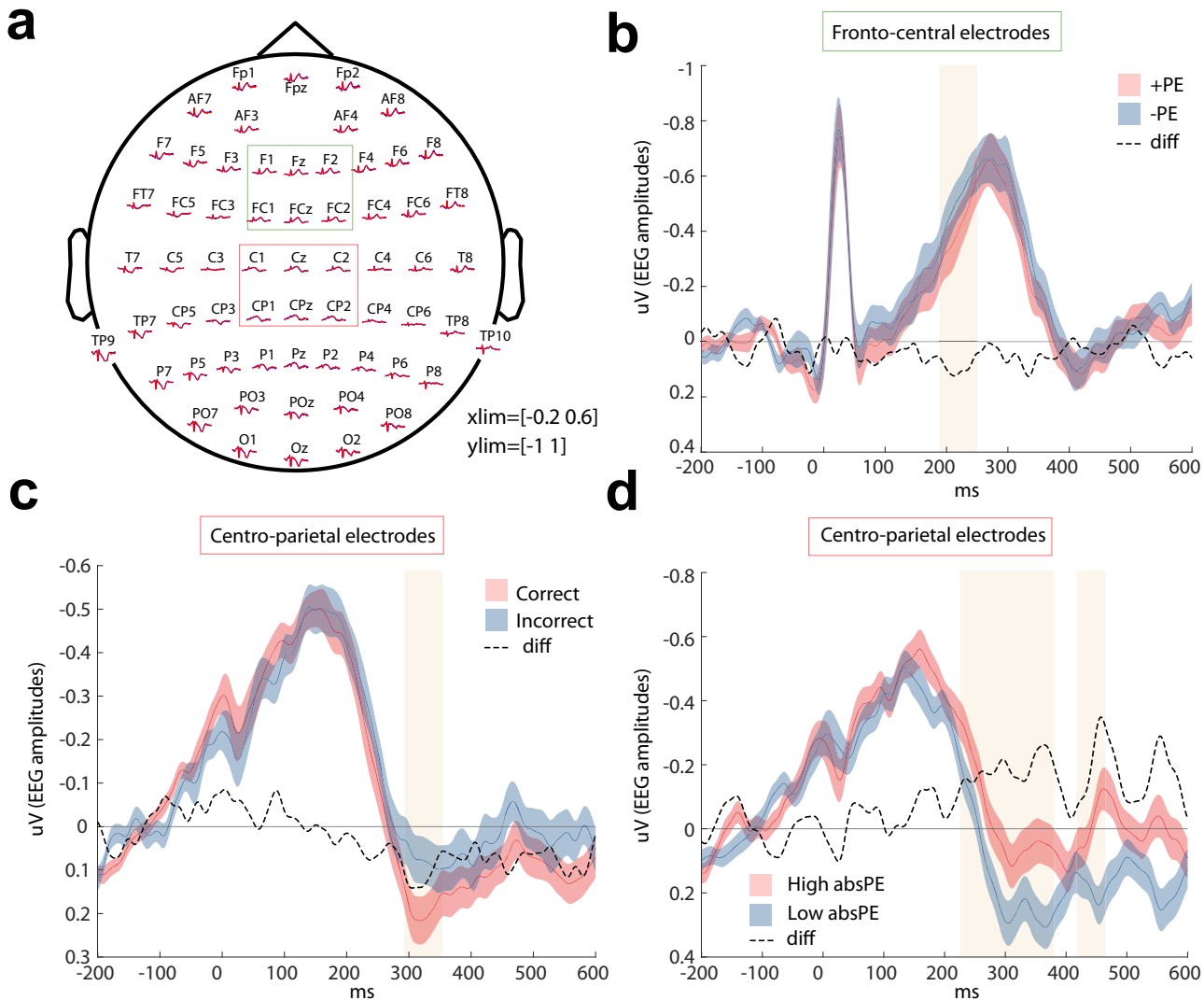

**Fig. 2 | HEP morphology and results. a** Grand average waveforms across the scalp time-locked to the onset of the R-wave which is the biggest electrical wave generated during normal conduction (time 0 ms, see "Methods" section). The set of electrodes clustered by ROIs (colour-coded) for the frontocentral and central-parietal electrodes are represented for further analyses (**b**–**d**) HEP waveforms across all trials following the onset of the R-wave (at time 0 ms) are shown separately (averaged all HEPs after feedback). This is presented for positive and negative signed prediction errors (PEs) for the frontal cluster in (**b**); correct vs incorrect outcome in (**c**); and for high and low surprising outcomes (absolute PEs) in (**d**). The dotted line represents the difference between the conditions (represented in red and blue). The shaded areas represent the time windows where there is a significant difference between conditions (*N* = 32 participants) revealed by cluster-based permutation analysis (two-sided).

The results of the cluster-based permutation analysis (see "Methods" section) revealed an increased HEP amplitude for trials with negative signed PEs in comparison to positive signed PEs (Monte–Carlo *p*-value = 0.004, Cohen D = 0.695) between 198 and 252 in the frontocentral sites (Fig. 2b). We also found a significant difference between correct and incorrect outcomes at a cluster around 250 ms after feedback (Fig. 2c, Cohen D = 0.696) in the same cluster. When contrasting trials in the absolute PE domain, we found multiple time point with significant difference in the HEP amplitude between the high surprising trials as opposed to the low surprising trials (Monte–Carlo *p*-values < 0.003, Cohen D = −0.724 cluster 1 and Cohen D = −1.17 cluster 2); these differences were observed in two clusters: cluster 1 with latencies 252–292 ms where a greater HEP amplitude for high vs low surprising trials was observed, and a cluster between 418 and 464 ms where the negative HEP deflection exhibited a greater amplitude for low vs high surprising trials. Both clusters were observed in the centro-parietal sites (Fig. 2d).

## The HEP is related to trial-by-trial variation in the absolute PE dimension

Rather than inspecting specific electrode averages, we next wanted to search for EEG heartbeat features that predict the learning axes. We thus moved on to identify the whole-brain heartbeat-evoked neural components of learning by using a multivariate single-trial discriminant analysis of the EEG (regularised Fisher Discriminant analysis, see "Methods" section) of the HEP-locked signals. More specifically, for each participant, we used the average of the HEP for each outcome (see Fig. 3a) and calculated the linear weights associated with each electrode that maximally separated (1) positive and negative signed PEs (Fig. 3e) and (2) high versus low magnitude of absolute PE (i.e., the size of the unsigned PE which describes how surprising the outcome is) (Fig. 3a). We did this over multiple temporal windows and quantified the classification performance by using the area under the curve (Az) using a leave-one out approach. This method has been well established in EEG data analysis[13,38,39].

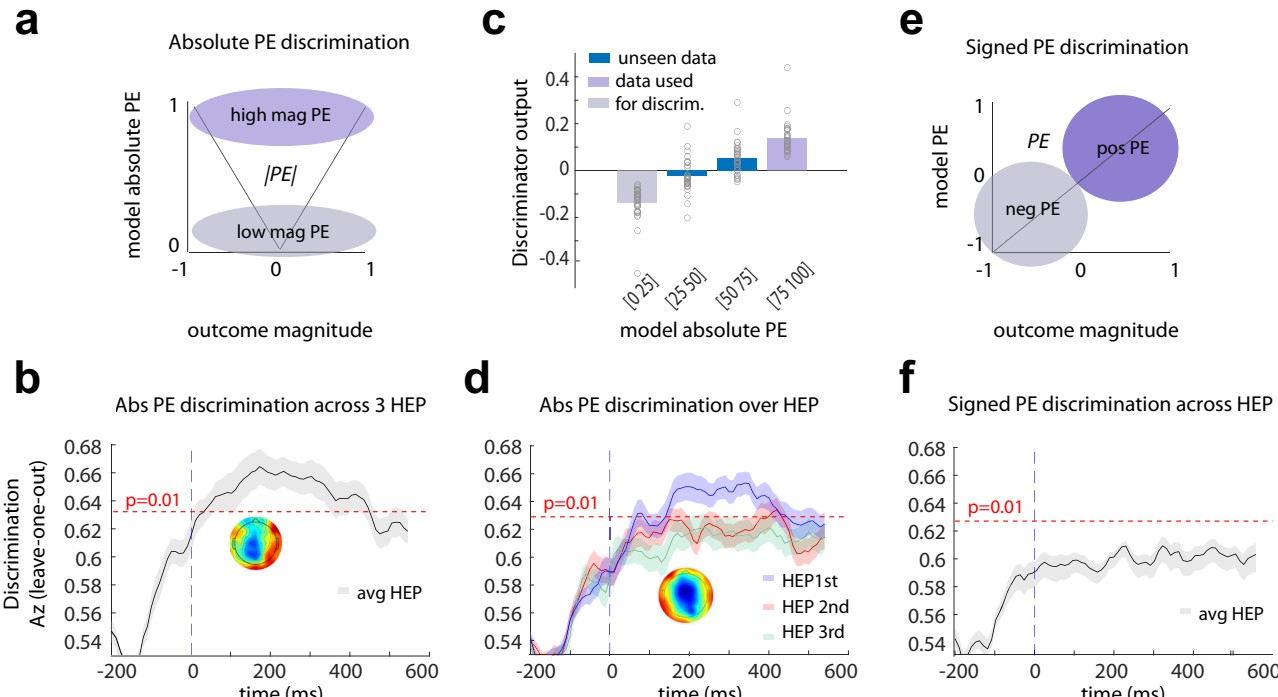

**Fig. 3 | Machine learning discrimination. a** Description of the data used for the absolute prediction error (PE) discrimination at time of outcome: we used the highest and lowest quantiles based on absolute PE (salience or surprise) as estimates for high and low absolute PE respectively. The analysis was performed on the HEP-locked EEG data. **b** Discriminator performance (cross-validated Area under the curve, Az) during absolute PE discrimination (low vs. high surprising outcomes), averaged across subjects ($N = 32$ participants). The dotted line represents the average Az value leading to a significance level of $P = 0.01$, estimated using a bootstrap test (two-sided). Shaded error bars are standard errors across subjects. This analysis used all first three HEP after outcomes. The scalp map represents the spatial topography of the absolute PE component. **c** Mean discriminator output (y) for the absolute PE component, binned in four quantiles based on model-based absolute PE estimates, showing a parametric response along the absolute PE dimension. Purple bins indicate trials used to train the classifier, while blue bins contain "unseen" data with intermediate absolute PE levels. Points are individual subjects. **d** The same analysis as in Fig. 3b ($N = 32$ participants) for each HEP after outcome. Blue represents the first heartbeat after outcome, red the second and green the third. The dotted line represents the average Az value leading to a significance level of $P = 0.01$, estimated using a bootstrap test (two-sided). Shaded error bars are standard errors across subjects. The scalp map represents the spatial topography of the absolute PE component. **e** Description of the data used for the signed PE discrimination: we used outcomes defined by the RL model as either positive or negative PEs. **f** Discriminator performance (cross-validated Az) during signed PE discrimination (positive versus negative PE), across all HEP after outcome, averaged across subjects ($N = 32$ participants). The dotted line represents the average Az value leading to a significance level of $P = 0.01$, estimated using a bootstrap test (two-sided). Shaded error bars are standard errors across subjects. No components were identified.

Using this machine learning approach, we showed the presence of a large heart-related component reliably discriminating – even in individual participants (see Supplementary Fig. 2) – between very high versus very low absolute PE outcomes. This component peaked in the time range 100–300 milliseconds after the *R*-wave (see EEG analysis for *R*-wave definition; Fig. 3a, b). On the other hand, we did not observe any heart-related component discriminating between positive and negative PEs (Fig. 3f). By contrast, as noted, a grand average response difference between positive and negative signed PEs was identified in the ERP in frontocentral electrodes (Fig. 2b). In conjunction, the two results suggest that there is, on average, a difference in cardiac-related signals when the valence of the signed PE is positive or negative but that trial-by-trial variability in the HEP does not statistically covary on a trial-by-trial basis with the trial-by-trial change in signed PE (Fig. 3f).

In summary, the different analysis approaches (ERP and machine learning) suggest the possibility of a number of relationships between HEP and learning signals but converge in suggesting an especially clear link, even at the trial-by-trial level, between the HEP and absolute PEs connoting how surprising or salient an outcome is. We therefore focussed our analysis on the HEP component carrying absolute PE information that we refer to as absPE-HEP. It is important to note that the regularised Fisher discriminant analysis and the mass-univariate analysis rely on fundamentally distinctive features of EEG data. The

mass-univariate analysis focuses on discriminating amplitude changes of event-related potentials resulting from averaging all trials in a few electrodes. By contrast, the machine learning approach capitalises on the trial-to-trial variability of the EEG data computed across all the recording sites which allows us to accurately measure the changes in the HEP signal that fluctuate with time on a trial-by-trial basis whilst learning is taking place.

To test whether the heart-related absolute PE (absPE-HEP) was parametrically modulated by the model absolute PE estimated in our model (rather than responding categorically to very high vs. very low absolute PE), we then calculated the discriminator amplitudes for trials with intermediate absolute PE levels (i.e. low absolute PE [0.25–0.50]; and high absolute PE [0.5–0.75]) which were not originally used to train the classifier (also called the "unseen" data). To do so, we applied the spatial weights of the peak discrimination performance for the extreme outcome absolute PE levels to the EEG data with intermediate values. We expected that the discriminator amplitudes for these previously "unseen" trials would increase linearly as a function of absolute PE. Thus, the resulting mean amplitude at the time of peak discrimination would proceed from very low < low < medium < very high absolute PE. This is indeed what we found (Fig. 3c, blue: intermediate categories, grey: categories used for discrimination) confirming the linear relationship between the absPE-HEP component and its model-based counterpart (test on the left-out data: $t_{31} = -7.303$; $p < 0.001$;

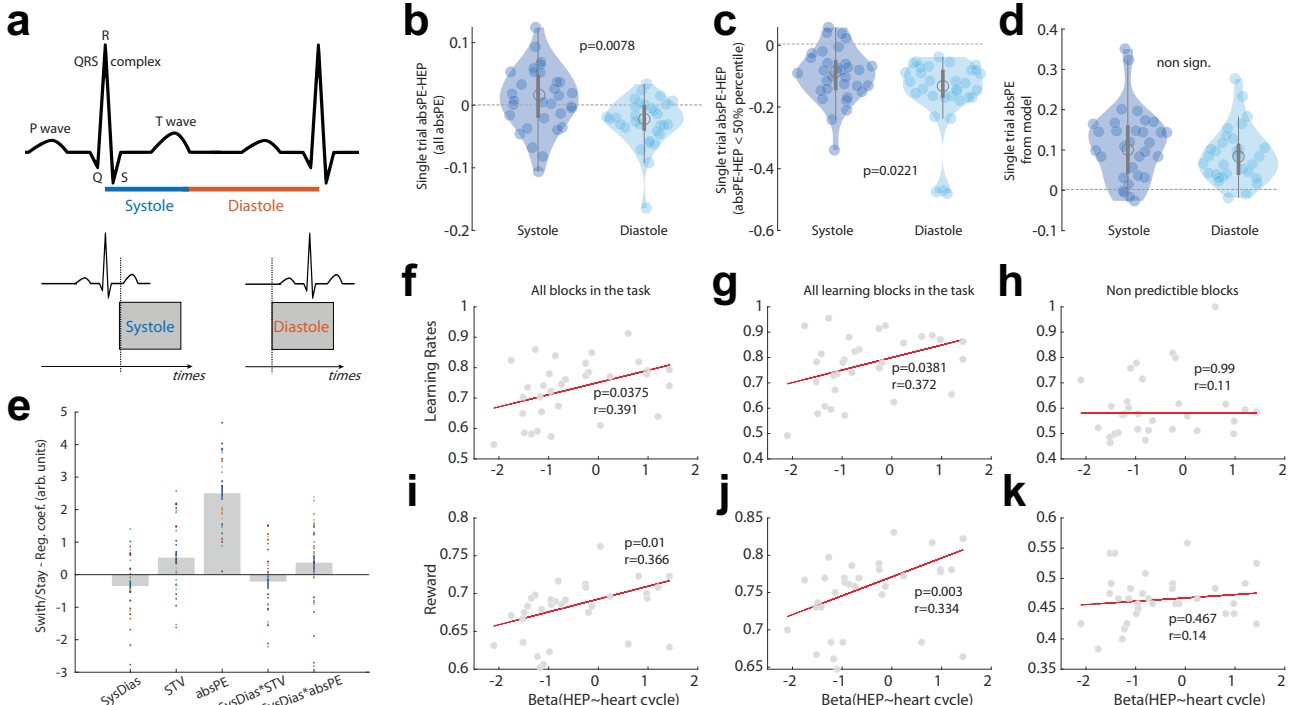

**Fig. 4 | Influence of the cardiac cycle on the absPE-HEP and learning.**
**a** Schematic description of the systole and diastole phases. In red and blue are the systole and diastole periods respectively. Below is a representation of two example trials on which outcome onsets happened at systole or diastole. We then looked at the EEG response locked to the next heartbeat. **b** Mean amplitude difference between the absPE-HEP of the first heartbeat after all outcomes presented at systole versus diastole ($N = 32$). A violin plot is used to present all individual participants' averages, as well as the mean and SEM. **c** Mean amplitude difference in the absPE-HEP for the low salient outcomes presented at systole versus diastole across subjects ($N = 32$ participants). **d** Mean amplitude difference in absPE-HEP for the high salient outcomes presented at systole versus diastole ($N = 32$ participants) – in **b**, **c** data are presented as mean values ± SE **e** A logistic regression analysis showed that switch/stay (1/0) could be predicted by several predictors, including the

Absolute PEs (participants were more likely to switch after a highly surprising outcome) but also residual absPE-HEP (predictors depicted from left to right: Systole/Diastole, Single-trial variability, Absolute PE, Systole/Diastole by Single-trial variability, Systole/Diastole by Absolute PE) **f**–**k** Results of the correlation between the regression coefficient for each participant between absPE-HEP and systole/diastole and the mean reward and learning rates in the task. In red, the fit of the robust regression. Any of these results remain true even when including a covariate indexing features of the external outcome type – reward and absolute PE from the model. Particularly, in **f** learning rates – all task blocks **g** learning rates – predictive blocks **h** learning rates – non-predictive blocks **i** reward – all task blocks **j** reward – predictive blocks **k** reward – non-predictive blocks. We did not adjust the $p$-value for multiple comparison.

CI = [−0.118 −0.066], Cohen D = −1.129) and also the generalisability and robustness of our machine learning approach. Having applied the estimated electrode weights to single-trial data to produce a measurement of the discriminating component amplitudes (representing the distance of individual trials from the discriminating hyperplane), we thereafter used these amplitudes for all subsequent analyses involving absPE-HEP.

Thus far, we have demonstrated that the largest the temporal component locked at $R$-wave onset, the biggest wave generated during normal heart conduction (see "Methods" section for full definition) for all heart-evoked signals collected during the feedback period linked with the absolute PE. However, the extent to which this temporal component was driven by the first, second, or third heartbeat that occurred in the outcome period remained unknown. Next, we therefore repeated our multivariate analysis independently for each of the three possible HEP times post-outcome, sequentially, across all trials to better understand the temporal dynamics of the absPE-HEP modulation. This approach allowed us to determine which heartbeat was most related to absolute PE. Applying this method, we showed that only the first HEP after feedback contained information about absolute PE that could be revealed with machine learning techniques, in the range 100–300 milliseconds after heartbeat onset (Fig. 3d). This finding indicates that the first heartbeat after outcome is the one that relates most to the representation of the absolute PE of the outcome, suggesting that the timing of the outcome with respect to the cardiac

cycle might be important in determining how participants update their internal representations of decision outcomes. Because our results highlight the importance of considering the first HEP after feedback rather than averaging all HEP after feedback, we also decided to redo our initial ERP analyses using only the first HEP after feedback albeit a lower statistical power. This is presented in Supplementary Fig. 5.

**Effect of the cardiac cycle timing on the absPE-HEP amplitude**
Having identified an HEP component associated with absolute PE (absPE-HEP), we next asked whether the timing of the outcome along the cardiac cycle further modulated the amplitude of this signal. In other words, we examined whether the magnitude of the absPE-HEP in the EEG epoch related to the first heartbeat after feedback onset would be higher or lower when the outcome was presented during diastole compared to systole. This would be an indication that internal subjective representations of how surprising an outcome is (compared to expectation), depend on the natural oscillation of the heart. To answer this question, we identified all outcomes with onsets which happened at diastole and all outcomes with onsets which happened at systole (see Fig. 4a). This split allowed us to compute the mean absPE-HEP for heartbeats after outcomes presented at diastole, and after outcomes presented at systole. Importantly, although the outcome onset was defined according to the systole and diastole periods in the previous $R$-wave, the absPE-HEP component that we analysed were defined in the closest $R$-wave (see "Methods" section). Naturally, as the diastole phase

is longer on average than the systole phase, we expect a higher number of outcomes presented during the diastole phase ($m = 65 \pm 5$ and $54 \pm 5$ – N: 32, mean and SD for diastole and systole, respectively). We also found that all other aspects of the task were not statistically different in these conditions. The frequency of occurrence of both conditions was not statistically different in the different learning blocks employed in the task (e.g. predictive or non-predictive; see "Methods" section; systole $F_3 = 0.2$, $p = 0.893$, $\eta^2 = 0.005$; diastole $F_3 = 0.2$, $p = 0.896$, $\eta^2 = 0.005$, Supplementary Fig. 3a). The two phases were also associated with levels of overall reward received in the task that were not statistically different ($t_{31} = 0.8046$, p = 0.427, CI = [−0.008 0.02], Cohen D = 0.14, Supplementary Fig. 3b), unsigned PEs from the RL model ($t_{31} = −0.058$, $p = 0.954$, CI = [−0.026 0.025], Cohen D = −0.01, Fig. 4d) or signed PE ($t_{31} = −0.72$, $p = 0.477$, CI = [−0.042 0.02], Cohen D = −0.12).

Having split the outcomes according to the cardiac cycle, we then moved on to test whether the associated absPE-HEP depended on whether the outcome was presented at systole or diastole. This is indeed what we found. The mean absPE-HEP was more negative when outcomes were presented at the diastole compared to the systole phase (t-test: $t_{31} = 2.8460$, $p = 0.007$, CI = [0.0107 0.065], Cohen D = 0.55, Fig. 4b). Similarly, we tested, with a mixed effects linear model, whether the STV could be predicted by a more complex model, including the trial-by-trial model-based absolute PE which is expected to covary with STV, the heart cycle (categorical variable) and the interaction term. Beyond the linear relationship between absolute PE and single-trial HEP which is to be expected (also see Fig. 3c) - such that this relationship increased as absolute PEs became smaller (main effect of absPE in mixed-effect model: $t_{122} = 3.539$, $p < 0.001$, 95% CI, [0.11 0.39], Partial Eta$^2$ = 0.644), we also found a main effect of cardiac cycle on the STV (main effect of heart cycle in mixed-effect model: $t_{122} = −2.336$, $p = 0.021$, 95% CI, [−0.35 −0.03], Partial Eta$^2$ = 0.05; interaction effect: heart cycle*absolute PE: $t_{122} = 1.96$, p = 0.052, 95% CI, [−0.0008 0.2], Partial Eta$^2$ = 0.035). Another analysis, independent of the previous one, showed the difference between the STV absPE-HEP < 50% percentile for systole and systole (Fig. 4c). This confirms that timing within the cardiac cycle modulates neural signals of absolute PE and that these representations are stronger after an outcome is presented at diastole compared to systole.

We then decided to investigate the relationship between information provided to participants on single trials and the cardiac cycle to test the idea that it should be possible to see a link between trial-by-trial variation in absPE-HEP and trial-by trial variation in updating of the values estimated for each choice. To do this we investigated whether the absPE-HEP at outcome (t) could predict change in choices in the next trial at $t + 1$. To do so, we ran an additional mixed-effect model to predict choice switching behaviours at $t + 1$ with the residual absPE-HEP variance at t (after controlling for the part of the absPE-HEP that was collinear with the absolute PE) and added the systole and diastole as a separate regressor. Our results showed that participants were more likely to switch after a highly surprising outcome (see Supplementary Table 2, results for AbsPE) aligning with previous reports[13] but also that higher residual variance in absPE-HEP, even after controlling for that part of the absPE-HEP that was collinear with absolute PE, also predicted switch behaviours on the next trials (see Supplementary Table 2, results for STV). In addition, while the relationship between the cardiac cycle on the trials (t) (whether the outcome's onset happened at systole or diastole) and switch at ($t + 1$) did not reach conventional significance (results for SysDias: p = 0.059, see Supplementary Table 2), the interaction term between SySDias and absPE was significant ($p = 0.035$, see Supplementary Table 2, SysDias: AbsPE). This indicates, that, as absPE decreases, the impact of the cardiac cycle on the next trial increases. The GLM is presented on Fig. 4E and all statistics reported in Supplementary Table 2.

We then asked whether these differences in the way naturally occurring bodily oscillations modulate the neural signals that determine learning can also mediate participants' decisions (which should be guided by learning that is based on the same neural signals). As we had found a more negative absPE-HEP when outcomes were presented at diastole compared to systole, we wondered whether interindividual differences in the link between the cardiac cycle and the absPE-HEP component co-varied with task performance and learning. We thus first ran a regression analysis for each participant to test the extent to which the cardiac cycle influenced their neural activity. Participants with a higher regression coefficient would have a stronger decrease in the absPE-HEP for outcomes presented at diastole compared to systole (see Fig. 4f–k). We expect these participants to be the ones showing a greater propensity for learning as their sensitivity to near-threshold events would be enhanced. These should also be the ones who ultimately receive more rewards overall. To test this hypothesis, in a second step, we ran a correlation between the regression coefficient and the mean reward and learning rates in the task. In line with our predictions, we found that participants showing a higher difference in absPE-HEP between diastole and systole were the participants that had higher learning rates and better task performance as indexed by the total number of rewards received (learning rates: $t_{30} = 2.176$, $p = 0.037$, Pearson $r = 0.391$; reward: $t_{30} = 2.74$, $p = 0.01$, Pearson $r = 0.366$; Fig. 4f, i). In summary, we can link interindividual variation in cardiac modulation of learning signals to interindividual variation in the parameters of a computational model of learning and to individual variation in an index of behaviour – overall rewards – independent of the computational model. To further examine the relationship between diastole-based absPE-HEP-related neural indices and learning, we examined task blocks where learning was possible (predictive blocks) or not possible (non-predictive blocks). The relationship was only present in the blocks in which learning was possible (predictive blocks: learning rates: $t_{30} = 2.17$, p = 0.038; Pearson $r = 0.372$; reward: $t_{30} = 3.21$, $p = 0.003$, Pearson $r = 0.334$; Fig. 4g, j; non-predictive blocks: learning rates: $t_{30} = −0.006$, p = 0.99; Pearson $r = 0.116$; reward: $t_{30} = 0.736$, $p = 0.467$, Pearson $r = 0.14$; Fig. 4h, k). These results remain true even when including a covariate indexing features of the external outcome type – reward and absolute PE from the model as opposed to the internal, subjective, absolute prediction error represented in the absPE-HEP (see Supplementary Fig. 4).

## Discussion

In this study, we combined machine learning-based analysis techniques and EEG to investigate the contribution of cardiac-related neural signals on several dimensions of reward-based learning. Our results demonstrate that the HEP recorded during the presentation of reward-related outcomes discriminates between different levels of absolute PE outcome. By contrast, the magnitude of the HEP was not statistically different when contrasting positive versus negative signed PEs. The absolute PE and signed PE components of reward learning subserve different functional roles in learning[13,16,33]; whilst signed PE is associated with approach-avoidance behaviour, absolute PE, also called salience impacts on future attentional engagement; an effect that is determined by the magnitude of the discrepancy between prior expectations and outcome. During learning, we found that single-trial HEP sizes were also related to absolute PE sizes when decision outcomes occurred. Moreover, some of the variation in the absPE-HEP also predicted whether participants were more likely to shift to a different decision when the decision they had just made had led to a surprising outcome. In other words, the absPE-HEP did not predispose participants to make one choice or another but larger absPE-HEP were associated with surprising feedback information and this was linked to learning.

The relationship between the cardiac cycle and learning is analogous to some cardiac-related effects that have been reported in the

context of decision-making. For example, cardiac responses in decision-related brain areas such as ventromedial prefrontal cortex are larger when the decision-related information will have a bigger impact on the decisions made[4]. Both in the decision-making results previously reported and in the current study, neural responses to the cardiac cycle are related to how impactful concurrent information will be on behaviour rather than with a particular type of behaviour. Similarly, single neuron responses recorded in adjacent orbitofrontal and anterior cingulate brain areas in macaques also vary with heart rate and heart rate is associated with a general increase in the speed of decision-making[5].

Cardiac neurophysiological responses often convey not only information about the current bodily state, but they also carry predictions of how the bodily system should organise internal resources to deal with expected future sensory information[40,41]. These cardiac predictions are often accompanied by a modulation of attentional responses to upcoming stimuli that, ultimately, are homeostatically relevant. In this way, it has been suggested that the internal bodily state determines perceptual stimulus salience in relation to homoeostatic levels[40,42]. For example, a stimulus occurring when resources are sparser may be perceived as more salient than a stimulus occurring when more resources are available. The absPE-HEP might signal that more attention needs to be deployed to the current outcome given the current bodily state. In this way a bodily signal might modulate learning.

Neuronal models of interoception conceptualise cardiac predictions as afferent signals projecting to agranular visceromotor areas in frontal cortex and anterior insula cortex, which serves as the primary interoceptive cortex[43,44]. The anterior insula is argued to be a main neural source for the HEP along with other interconnected areas such as the cingulate and the somatosensory cortices[45–47]. These brain regions belong to a wider network, often referred to as the salience network, which is sensitive to homeostatically relevant stimuli independent of whether their valence is negative (penalising) or positive (reinforcing)[31]. It is becoming increasingly clear that neural responses in the absolute PE network rise quickly after an outcome is revealed[30,38]. Here we observe that the HEP is parametrically modulated by the outcome's absolute PE and that this is mainly due to the first heartbeat recorded immediately after the outcome onset. This means that HEP magnitude changes recorded immediately after outcome can be used as a proxy for attentional allocation to the internal representation of absolute PE.

It is worth noting that our current results do not allow us to support the idea that cardiac deceleration – i.e., longer diastolic phases, serves to make the organism better equipped to intake sensory stimuli and respond to them[27]. Future studies should tackle this limitation and further investigate the precise relationship between trial-by-trial amplitude changes in the HEP and humans' ability to integrate sensory information after a positive or negative feedback.

Previous studies have carefully time-locked the presentation of stimuli to the cardiac phase to investigate differences in the way that stimuli are processed[2,6,21]. For example, tactile stimuli presented at diastole are more frequently detected than those presented at systole[2]. Conversely, the ability to control movements is facilitated during cardiac systole[8] – albeit this tendency reverses when emotional cues are present[9]. However, sensory or learning information is not presented in such a phase-locked manner in our everyday lives. By investigating how participants naturally receive information relevant for learning and assign credit for outcomes to objects maintained in memory, with respect to the natural timing of the cardiac cycle, we have adopted an ecological approach to studying brain-heart interactions in the context of learning and decision-making. Previous studies, adopting a similar approach, have shown that people actively seek information in the world, or more precisely sample the world through active sensing, as a function of the cardiac cycle. For example, in an active sampling visual paradigm, saccades and visual fixations are more likely to occur in the quiescent phase of the cardiac cycle (e.g. diastole)[48]. Similar work suggests that people actively adjust sensory sampling so that more time is spent in the diastole period in which perceptual sensory sensitivity is enhanced[49]. Moreover, in dyadic interactions actions are more likely to take place during diastole, and also the observer is less likely to experience a heartbeat (systolic phase) when observing movement endpoints[7]. In our study, we have shown that the magnitude of the single-trial HEP is stronger when the outcome appeared during the diastole period in comparison to the systole period (Fig. 4). This suggests that the phase of the cardiac cycle is an important modulator of internal representation and cognition and influences the way in which we naturally receive information.

Importantly, we also observed that the influence of the cardiac cycle on the absPE-HEP magnitude progressively increased as the outcome absolute PE became smaller. In outcomes with near-threshold absolute PEs, the absPE-HEP magnitude increase was predominantly observed when the outcome was presented at diastole (Fig. 4). This means that when the decision maker's prior expectations are close to the outcome (i.e., small adjustments between expectations and outcomes) learning is more likely to occur during the quiescent phase of the cardiac cycle than during the active, systolic phase. Neuronal excitability is influenced by the cardiac cycle; whilst neural signals from the baroreceptors occurring at systole attenuate concurrent brain activity[24,50] and impair information processing, enhanced excitability and perceptual processing is observed during diastole[2,20]. Formally, enhanced neuronal excitability may increase neural gain, which directly translates into an increase of the breadth of attention towards the aspects of the environment to which one is predisposed to attend[51]. Here we show that in instances where learning happens in small increments because the PE-related surprise is not very salient, learning is enhanced during diastole compared to systole, helping to update prior expectations even when there is little new information available.

Beyond showing modulations of the absPE-HEP amplitude timed to different phases of the cardiac cycle, our results demonstrate that these heart cycle-specific neuronal changes translate into individual differences in overall learning. Individuals that exhibited higher differences in the absPE-HEP magnitude changes to outcomes presented at diastole versus systole also showed higher learning rates and better overall task performance. Individual differences in cardiac neural responses have long been established[52]. For example, HEP amplitude modulation often present during observation of highly salient stimuli is stronger for individuals with greater self-reported empathy scores[53]. Also, individuals with low cardiac interoceptive sensitivity show greater difficulty retrieving information presented at systole in comparison to those with high interoceptive sensitivity[54]. Additionally, we found that these individual differences in the relationship between the cardiac cycle and absolute PE encoding were only true in task blocks where learning was taking place versus blocks where learning was precluded (i.e., random contingency between colours and stimuli). Increased and decreased cardiac sensitivity has also been shown to help or hinder adaptive intuitive decision-making when the generated cardiac predictions favour advantageous choices - i.e., when learning is taking place; however, the opposite is true when predictions are towards disadvantageous choices[55].

Our finding that absPE-HEP representation depends on the heart cycle might also be described in terms of periodical modulations of internal value representations in a predictive coding framework. According to this framework, the brain is constantly creating and updating predictive internal models of sensory inputs, including both exteroceptive and interoceptive signals such as the heartbeat. As each heartbeat and its accompanying pulse wave cause temporary physiological changes throughout the body, the brain treats these recurring cardiac signals as predictable events and attenuates them to reduce

the chances of mistaking these self-generated signals for external stimuli[56–58]. As a consequence, for example, in the context of somato-sensory events, sensory discrimination is less accurate during systole than diastole[59]. However, here, we have shown that even if the sensory information is the same, the extent to which an absolute PE affects learning is also linked to the cardiac cycle; the degree to which an internal model of a cue-outcome association is strengthened depends on the cardiac cycle, in line with a predictive coding account for cardiac phase-related internal fluctuations.

## Methods

This study was approved by the University of Oxford Medical Science Interdivisional Research Ethics Committee, Oxford RECC, No. R55856/RE002.

### Participants

Thirty-five healthy, right-handed adults participated in the experiment. Three participants were excluded due to excessive noise in the EEG signal so that data from 32 participants were included in the analyses ($24 \pm 7.13$; 10; $0.83 \pm 0.13$); where numbers correspond to mean age ± SD; number of female participants, handiness mean ± SD; as measured by the Edinburgh handedness inventory[60]. Participants gender was determined based on self-report and it was not considered as an experimental variable in our design because there is no prior evidence of gender difference in the processes investigated. All participants were naïve to the task, had no personal or familial history of neuro-logical or psychiatric disease, were right-handed, gave written informed consent (Medical Science Interdivisional Research Ethics Committee, Oxford RECC, No. R55856/RE002), and received monetary compensation for their participation. Sample sizes were determined based on previous studies that have used similar reward learning paradigms to investigate brain responses during learning[13,34,38] and studies that have measured the HEP to investigate neural responses to heartbeats in humans[2,61]. No statistical method was used to pre-determine sample size.

### Stimuli

Stimuli consisted of pictures of 10 faces and 10 houses ($512 \times 512$ pix-els), two circles in blue and orange ($125 \times 125$ pixels). All the stimuli were equalised for luminance and contrast. The outcome images consisted of a tick and a cross, which were also equalised for luminance and contrast. The face database was provided by the Max-Planck Institute for Biological Cybernetics in Tuebingen, Germany[62].

### Experimental design

Participants were seated in a dimly lit, sound-attenuated, and elec-trically shielded chamber in front of a monitor at a distance of 70 cm. EEG was recorded using a 64-channels cap (see "EEG data collection" section) while participants performed a reward-based learning task. Participants' ECG was recorded with a standard EEG electrode attached to their chest to monitor heart activity throughout the session. The experiment consisted of eight blocks of 60 trials (480 trials in total) separated by small breaks, following a repeated measures design. At the beginning of each block, the association between colours and objects changed. Two new objects were presented in each block: a house and a face. Each object was uniquely associated with a colour according to different schemes. There was a total of four types of blocks: three predictive blocks (in which objects predicted outcomes) and one non-predictive block (from which predictive associations between objects and outcomes were absent). The three predictive blocks contained the following associations: (1) both stimuli were highly predictive and there was a negative correlation between each stimulus and respective associated outcomes (i.e., each stimulus pre-dicted a different colour), (2) both stimuli were highly predictive and positively correlated in the outcomes that they predicted (i.e., both

stimuli predicted the same colour), (3) only one stimulus was highly predictive and the other non-predictive. In the non-predictive block, the two objects were not associated with any colours.

### Learning task

We used a modified version of the weather prediction task. In a typical version of the task, participants have to predict the weather (rain/sun) on the basis of probabilistic cues. To avoid any subjective preference, we changed the sun/rain to two neutral colours (light blue/orange). We also presented one object at a time to isolate the EEG responses to faces and houses. On each trial, participants first saw a fixation cross for 500 ms, followed by the presentation of one stimulus that could be either a face or a house (500 ms). This was repeated for the second object (same timing). Each possible pair of objects: Face-House, House-Face, Face-Face and House-House were presented equally often and counterbalanced across a block. After the presentation of both objects, participants had to make a decision between two colours, orange and blue on the basis of their estimates of the association between the objects and colours as well as on the basis of what the particular combination of objects would be likely to predict. For example, if the house predicted orange deterministically (100%), the face predicted blue and they were presented together, then there was a 50%/50% chance of getting a blue/orange. If, however, the face was presented twice, then the outcome was blue, 100% of the time. The decision phase lasted 1200 ms. After participants made their decisions, they saw the outcome of their choice for 4000 ms, which allowed us to record on average four heartbeats per outcome (mean: 4.58, std ±0.85 across trials). After the task, participants were given a debrief and paid £20 for their participation. They were told that they would receive a fixed payment for participation (£15 per hour) and an additional amount (up to a maximum of £5) based on the outcome of a random subset of trials selected at the end of the experiment (excluding 'lost' trials). No further details regarding the mapping between earned points and the final payoff were given to the subjects.

### Computational modelling

Four RL models were used:

**Simple Cues model and Conjunction models.** We built on a Rein-forcement Learning framework to implement our first two Models that computed a Prediction Variable *PV* either after summing up the equally weighted stimulus-outcome association strengths for each cue V1 and V2 that is updated after each cue is presented (SimpleCue Model) or after a Face-Face V1, Face-House V2 or House-House V3 (Conjunction Model) is presented. Any cue or pair of cues is updated such as:

$$V_{c,n+1} = V_{c,n} + \alpha * PE \tag{1}$$

where PE is the prediction error (Outcome − $PV_n$). Note that absolute PE does not explicitly update value estimates in this model. PV is then converted to a choice probability following the equation:

$$p = 1/(1 + e(\beta * (PV - 0.5) + \gamma * C_{n-1})) \tag{2}$$

where β is the inverse temperature, or exploration parameter, and γ represents the choice stickiness[34,63] (the degree to which choices are likely to simply be repeated from trial-to trial regardless of outcome). $C_{n-1}$ is the choice in the previous trial (orange choice coded as +1 and blue choice coded as −1). *V* is the item−outcome association strength of each item, *O* is the outcome in the current trial (orange outcome coded as +1 and blue outcome coded as 0), and α is the learning rate shared by both items. The subscript *n* represents the current trial, and $n + 1$ represents the updated trial. There are three free parameters in this model: the learning rate α, the exploration parameter β, and the choice stickiness factor γ.

**Recency weighting model.** This model is very similar in essence to the SimpleCue Model but also presents a recency weighting at the time of learning which updates the value estimate for the second of the two presented cues more than the first (RL_HEP_rw). In this model, the value of the most recently presented cue is more strongly updated than the first one as a function of an additional free parameter *trace*. This model has four free parameters.

**Dynamic learning rate model.** We implemented a model in which the learning rate scales with the slope of the smoothed |PE|. This model reprises Pearce-Hall's theory that surprise drives the acquisition of stochastic stimulus-outcome contingencies. In this new model, the smoothing of the unsigned |PE| (the degree of which is regulated by a free parameter *rho*) should render the inference process about whether a change has occurred in the environment more robust to any inherent task stochasticity. Moreover, an additional free parameter *gamma* controls the extent to which the dynamic updating of the learning rate is influenced by the slope. For example, whilst lower values of *gamma* yield substantial trial-by-trial changes of the dynamic learning rate even in the presence of small slope estimates (that is, low surprise), higher values of *gamma* result in a more stable learning rate even in the presence of significant slope estimates (that is, high surprise). Hence, this model also allows for the possibility that subjects might be employing a relatively fixed learning rate.

**Model fitting.** All RL modelling was conducted in Matlab (version 2020a). We used an iterative expectation-maximization (EM) algorithm as in previous work[63] to fit the models. During the expectation procedure, we computed the maximum posterior likelihood ($NPL_i$) calculated with the parameter vector $h_i$ of each block $i$ ($i \in \{1..N\}$), given the choices and group-level Gaussian distributions over the parameters (mean vector $mu$ and standard deviations $sigma$) as per the following:

$$NPL_i = \max_h \left[ \sum_t (\log(p_t(choice|h)) + \sum_{mu,sigma} \log(normpdf(h|mu,sigma)) \right] \quad (3)$$

$$h_i = argmax_h \left[ \sum_t (\log(p_t(choice|h)) + \sum_{mu,sigma} \log(normpdf(h|mu,sigma)) \right] \quad (4)$$

The first part of the equation describes the likelihood of the observed choices given a vector of free parameters and the second part captures the likelihood of these parameters given a normal group-level distribution. We initialised the group-level Gaussians as uninformative priors with means of 0.1 (plus some added noise) and variance of 100. During the maximisation step, we recomputed $mu$ and $sigma$ based on the estimated set of $h_i$ and their Hessian matrix $H_i$ (as calculated with Matlab's *fminunc*) overall $N$ sessions.

$$mu = \frac{1}{N} \sum_i h_i \quad (5)$$

$$sigma^2 = \frac{1}{N} \sum_i \left[ h_i^2 + diag(pinv(H_i)) \right] - mu^2 \quad (6)$$

where the diagonal terms of the inverted Hessian matrix (computed in Matlab with *diag(pinv(H_i))*) give the second moment around $h_i$, approximating the variance, and thus the inverse of the uncertainty with which the parameter can be estimated.

We repeated expectation and maximisation steps iteratively until convergence of the posterior likelihood $NPL_i$ summed over the group or a maximum of 800 steps. Convergence was defined as a change in $NPL_i < 0.001$ from one iteration to the next.

**Model comparison.** We compared fitted models by calculating their integrated BIC ($BIC_{int}$)[28,63,64]. For this, we drew $k = 1000$ samples of parameter vector $h_i$ per session $i$ from the Gaussian population distributions using the final estimates of $mu$ and $sigma$, and computed the negative log-likelihood ($NLL_{i,k}$) of each sample and session using the equation (corresponding to the first part in equation 1).

$$NLL_{i,k} = - \sum_t \log(p_t(choice|h_{i,k})) \quad (7)$$

Next, we integrated the $NLL_{i,k}$ over samples $k$ and sessions $i$ and calculated $BIC_{int}$ based on the integrated log-likelihood ($iLog$) in the following way:

$$iLog = \sum_i \log \left( \sum_{k=1}^{2000} e^{-NLL_{i,k}}/2000 \right) \quad (8)$$

$$BIC_{int} = - 2 \times iLog + Np \times \log \left( \sum_i Nt_i \right) \quad (9)$$

$Np$ refers to the number of free parameters per model and $Nt_i$ refers to the number of trials per session $i$.

As a second index of model fit, we used the Laplace approximation to calculate the log model evidence ($LME$) per session $i$ based on $NPL_i$ (see equation 1):

$$LME_i = - NPL_i - \frac{1}{2} \log(\det(H_i)) + \frac{Np}{2} log(2\pi) \quad (10)$$

We submitted the $LME$ scores to spm_BMS[65] to compute the 'exceedance probability', the posterior probability that one model is the most likely model used by the population among a given set of models. In addition, we computed the session-wise difference in $LME$ between two candidate models (the best and second best) to approximate log Bayes factors, i.e. the ratio of posterior probability of the models given the data.

**Parameter recovery**

We used the equations 1 and 2 with the fitted parameter to create synthetic choices based on $p$ (probability of choice), with a simple rule: for $p < 0.5$, the choice would be orange choice and for $p > 0.5$ the choice would be blue. We then fitted the model to the synthetic data.

**EEG and ECG recording**

EEG was recorded with sintered Ag/AgCl electrodes from 62 scalp electrodes mounted equidistantly on an elastic electrode cap (64Ch-Standard-BrainCap for TMS with Multitrodes; EasyCap; two cap sizes, 56 cm and 58 cm head circumference). The distance between electrodes was on average 3.3 cm and 3.5 cm for the 56 cm and the 58 cm cap, respectively. The Ground electrode was located centrally at the electrode site corresponding to AFz in the 10/20 system. An additional ECG electrode was placed on the participants' chests around 12 cm below the left clavicle. All electrodes were referenced to the right mastoid and re-referenced to the arithmetic average reference of all electrodes off-line. Continuous EEG was recorded using BrainAmp amplifiers (BrainProducts, Munich, Germany; 0.1 μV analogue-to-digital conversion resolution; 1000 Hz sampling rate; 0.01-100 Hz online cut-off filters).

## EEG data analysis

Off-line EEG analysis was performed using Fieldtrip (https://www.fieldtriptoolbox.org/). The data was digitally band-pass-filtered between 0.5 and 40 Hz. Bad/missing channels were restored using a FieldTrip-based spline interpolation (1–2 electrodes per participant on average). Detection of $R$-peaks in the ECG recording was done using the Pan-Tompkins algorithm as implemented in MATLAB[66]. Next, the data were segmented into intervals time-locked to either the onset of the feedback, or the R-peak onset of the heartbeat $R$-waves occurring during the feedback period, or the onset of the visual images (faces/houses). The $R$-wave is the biggest wave (indicating the changing direction of the electrical stimulus as it passes through the heart's conduction system) generated during normal conduction and the first upward deflection after the $P$-wave part of the QRS complex as presented in Fig. 4a. The $R$-peak of the $R$-wave determines the time 0 ms of our HEP.

The intervals time-locked to the feedback onset were segmented into 4.9 s intervals starting from 0.9 s before the feedback onset. The intervals time-locked to the onset of the $R$-wave were segmented into 0.8 s intervals starting from 0.2 s before the $R$-wave onset. The intervals time-locked to the onset of the visual images were segmented into 0.7 s intervals starting from 0.2 s before the stimulus onset. This was done separately for positive versus negative PEs, high versus low absolute PE, and for correct versus incorrect trials.

Automatic artefact rejection was performed excluding trials and channels whose variance (z scores) across the experimental session exceeded a threshold of 20 µV. This was combined with visual inspection for all participants eliminating large technical and movement-related artefacts. Physiological artefacts such as eye blinks, saccades and the volume-conducted cardiac-field artefact (CFA) were corrected, in all participants, by means of independent component analysis (RUNICA, logistic Infomax algorithm) as implemented in the FieldTrip toolbox. Importantly, the data could also be contaminated by stereotyped movement of tissue or sensor with the pulsed blood flow, which becomes averaged together into a voltage change. These artefacts would likely arise due to the motion of EEG electrodes as a result of local pulsatile movement of the scalp during the cardiac cycle, because of varying blood flow through scalp vessels during cardiac rhythms. Stereotyped movements of tissue or the sensor with the pulsed blood flow of the BCG artefact are characterised by their temporal relation with the cardiac rhythms captured by electrocardiogram. Several methods have been used to deal with these artefacts but one of the most successful methods is, in fact, ICA removal[13,38,67]. Those independent components (4.78 on average across participants; 1.13 SD) whose timing and topography resembled the characteristics of the physiological artefacts were removed. The CFA represents a challenge to the analysis of the HEP because the averaging of the data around the R-peak amplifies the CFA that are time-locked to the heartbeat[68]. Nonetheless, ICA has been shown to be of high efficiency in the removal of the independent components representing CFAs from the EEG signal[47,69–71]. The IC identification and selection process were guided by visual inspection of their properties, based on time course and scalp topography. ECG channels were excluded from the analysis and the signal was then re-referenced to the arithmetic average of all electrodes.

For the ERP analysis, the segments were baseline-corrected using an interval from −0.15 s to −0.05 s for the segments time-locked to the $R$-wave onset, an interval from −0.9 s to −0.1 s for the segments time-locked to feedback, and interval from −0.2 s to −0.05 s for the segments time-locked to the visual stimulus onset. To further ensure that the HEP changes that we observe are not influenced by CFA artefacts, and they are truly locked to the participants' heartbeat, we created surrogate $R$-peaks by shifting the onset of the original $R$-peak[45,47]. $R$-peaks were shifted within a time window of −500 to +500 ms and they were shifted by the same amount separately for each subject and each of the four learning blocks. We subsequently applied the same criteria for calculating HEP amplitude and submitted these surrogate values to the cluster-based permutation test as described below.

## Topography and statistical analysis of the ERPs

In light of the considerable variability in the polarity, latency and scalp distribution of the HEP [9, 10] we adopted a non-parametric, cluster-based permutation approach to first determine the HEP morphology, and then estimate any HEP amplitude modulation as a function of learning. Subject-wise activation time-courses were extracted and passed to the statistical analysis procedure in FieldTrip, the details of which are described by Maris and Oostenveld[72,73]; Subject-wise activation time-courses were compared to identify statistically significant clusters in the time and spatial domain using a FieldTrip-based analysis across all time points and electrode sites. FieldTrip uses a non-parametric method[73] to address the multiple comparison problem. $T$-values of adjacent temporal and frequency points whose p-values were less than 0.05 were clustered by adding their t-values, and this cumulative statistic is used for inferential statistics at the cluster level. This procedure, i.e., the calculation of $t$-values at each temporal point followed by clustering of adjacent t-values, was repeated 5000 times, with randomised swapping and resampling of the subject-wise time-frequency activity before each repetition. This Monte–Carlo method results in a non-parametric estimate of the $P$-value representing the statistical significance of the identified cluster.

The topographical distribution of the neural phenomena comprising the HEP was defined by computing mean voltages of the HEP time-locked to $R$-wave onset for all trials at the group-level using the cluster-based permutation test (one-tailed test) including all electrodes sites and across the entire time window where the HEP typically takes place, this is, 0.1–0.5 s[41,61,74,75]. In this analysis, no a-priori electrode clusters were formed (all active electrodes were treated as a distinct variable); one-tailed test was used to allow the contrast between the mean voltage of the HEP for all trials against zero. The topography analysis revealed a number of electrodes widely spread along the frontal, centro-frontal and posterior areas where the HEP was distributed. These electrodes were then organised in 2 ROIs, a frontocentral ROI and a centro-parietal ROI, according to their spatial distribution (Fig. 2a) for further processing.

Next, we used the cluster-based permutation approach as implemented in Fieldtrip (see below) to test if HEP varied across the two main dimensions of learning: signed and absolute PE as well as correct versus incorrect outcomes. Since this method allows the comparison of only two conditions, we first organised the trials in two categories. We thus computed averaged signals aggregating trials with positive PE *versus* negative PE; and trials with high absolute PE *versus* low absolute PE and trials with correct *versus* incorrect outcome. Thereafter, we ran three parallel contrasts on averaged HEP contrasting trials with correct *versus* incorrect valence; trials with high positive PE *versus* negative PE; and trials with high absolute PE *versus* low absolute PE, by means of within-subject non-parametric cluster-based permutation analysis as described above and represented in Fig. 2b. A non-parametric, cluster-based permutation approach is an efficient way of dealing with the multiple comparison problem that prevents biases in pre-selecting time-windows avoiding inflation of type I error rate. Thus, the statistical analyses were performed across the entire time window in which the HEP typically takes place (0.1–0.6 s) and restricted to the ROIs defined according to the HEP morphology analyses. This is a non-parametric test that does not assume normality of the data. Furthermore, given the repeated nature of the design (within-subject design), the variance between group comparisons should be comparable. For each comparison, subject-wise activations at electrode sites circumscribed in the ROI were extracted and passed to the analysis procedure. To avoid spurious findings, significant effects of 15 milliseconds or shorter were discarded from further analysis. Where appropriate,

p-values were corrected for multiple comparisons using Bonferroni-Holms correction.

## Multivariate analyses

We hypothesised that the HEP, that is, the epoched EEG data synchronised to the heart at time of outcomes, may be associated with reward-based learning dimensions. To investigate this idea, we used a linear multivariate classifier, with a sliding window approach, on the HEP data. Specifically, we searched for a projection of the multidimensional EEG signal, xi(t), where $i = 1…T$ and T is the total number of trials, within short time windows that achieved maximal discrimination between binary groups of trials as described in Fouragnan and colleagues[13,38]. Locked to the heartbeat, these binary groups included: (1) positive versus negative signed PEs, (2) very high and very low absolute PEs.

All analyses were performed on windows with a length of $N = 60$ ms and the window centre τ was shifted from −100 to 600 ms relative to the heartbeat onset, in 10-ms increments. We applied a regularised Fisher discriminant analysis to find the spatial weighting, w(τ), that maximally discriminated between the binary groups described above, arriving at a one-dimensional projection yi(τ), for each trial i and a given window τ:

$$y_i(\tau) = \frac{1}{N} \sum_{t=\tau-N/2}^{t=\tau+N/2} \boldsymbol{w}(\tau)^{\perp} \boldsymbol{x}_i(t) \tag{11}$$

where yi(τ), is organised as a vector of single-trial discriminator amplitudes (1 × Trials), the spatial filter, w(τ), is organised as a vector with as many weights as there are channels in the data (1 × 64) and data, xi(τ), is organised as a matrix, with dimensions (64 × Trials/Samples). We adopted this approach to identify all time windows τ yielding significant discrimination performance in the heart-related period. The projection vectors w at each time window τ were estimated as: w = Sc(m$_2$−m$_1$) where $m_i$ is the estimated mean of condition i and Sc = 1/2(S$_1$ + S$_2$) is the estimated common covariance matrix (that is, the average of the condition-wise empirical covariance matrices, with $T$ = number of trials). To treat potential estimation errors, we replaced the condition-wise covariance matrices with regularised versions of these matrices, with λ∈[0, 1] being the regularisation term and ν the average eigenvalue of the original Si (that is, trace(Si)/62). Note that λ = 0 yields unregularised estimation and λ = 1 assumes spherical covariance matrices. Here we optimised λ for each participant using a leave-one-out trial cross validation procedure across the entire post-outcome period.

We quantified the performance of the discriminator for each time window using the area under a receiver operating characteristic curve, referred to as an Az value, using a leave-one-out trial procedure. To assess the significance of the discriminator, we used a bootstrapping technique where we performed the leave-one-out test after randomising the trial labels. We repeated this randomisation procedure 1000 times to produce a probability distribution for Az (normally distributed) and estimated the Az leading to a significance level of $P < 0.01$.

Given the linearity of our model, we also computed scalp topographies of the discriminating components resulting from equation (1) by estimating a forward model as:

$$a(\boldsymbol{\tau}) = \frac{\boldsymbol{x}(\boldsymbol{\tau})\boldsymbol{y}(\boldsymbol{\tau})}{\boldsymbol{y}(\boldsymbol{\tau})^{\perp}\boldsymbol{y}(\boldsymbol{\tau})} \tag{12}$$

where yi(τ) is now shown as a vector y(τ), where each row is from trial i, and xi(τ) is organised as a matrix, x(τ), where rows are channels and columns are trials, all for time window τ. These forward models can be viewed as scalp plots and interpreted as the coupling between the discriminating components and the observed EEG.

## Diastole versus systole definition

Considering the biphasic nature of cardiac activity, we compared the cardiac neural response to absolute PE between the systolic and diastolic ventricular phases, namely, for simplicity, systole and diastole. We defined systole as the time between the $R$-peak and 300 ms after $R$-peak (to coincide with the end of $T$-wave) (Fig. 3a)[17,76]. We used the systolic offset of each cardiac cycle to define the onset of the diastole period, which ended at the $R$-peak. The non-equal length of systole and diastole meant that we were more likely (~60%) to have an outcome onset in the diastole phases of the cardiac cycle. Each outcome was categorised depending on whether the stimulus occurred during systole or diastole. The average number of trials categorised as systole was 54.57 and as diastole was 65.35 with standard deviation of 4.99 and 5.05 respectively. Importantly, when an outcome was assigned to systole or diastole, the assignment depended on that outcome's timing with respect to a current $R$-wave. However, the absolute PE-related HEP that were used for analysis in this work (Fig. 4) related to the next $R$-wave.

## Regression analysis

To examine the association between the cardiac cycle (i.e. diastole: 1, and systole: 0) and the neural cardiac-related signal, we performed the following logistic regression analysis (separately for each participant):

$$\text{HEP} \sim \beta 1 * \text{cardiac\_cycle} + (1|\text{subject}) \tag{13}$$

We then tested whether the regression coefficients across participants (β1 values in Eq. 3) came from a distribution with a mean different from zero (using a $t$-test). Data were tested for normality using a Kolmogorov−Smirnov test. To control for potential confound of outcome, we also performed the following logistic regression analysis (separately for each participant):

$$\text{HEP} \sim \beta 1 * \text{cardiac\_cycle} + \beta 2 * \text{outcome\_valence} + \beta 3 * \text{outcome\_surprise} + (1|\text{subject}) \tag{14}$$

We then tested whether the regression coefficients across participants (β1 values in Eq. 4) came from a distribution with a mean different from zero (using a two tailed t-test).

## Reporting summary

Further information on research design is available in the Nature Portfolio Reporting Summary linked to this article.

## Data availability

We have deposited all choice, EEG and ECG raw data in an OSF repository. All reinforcement learning results in this paper are derived from these data alone. The link to the data repository is: https://osf.io/qgw7h/ (https://doi.org/10.17605/OSF.IO/QGW7H). Source data are also provided as a Source Data file. Source data are provided with this paper.

## Code availability

The code of the reinforcement-modelling pipeline including model comparisons implemented in Matlab, as well as the Machine Learning code used on the EEG and the EEG preprocessing pipeline have been deposited in the following GitHub repository: https://github.com/efouragnan/EEG-CRS_learning. (https://doi.org/10.5281/zenodo.10370532).

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

## Acknowledgements

Funding for this work was provided by the UKRI and the BBSRC to Elsa Fouragnan (MR/T0223007/1 and BB/Y001494/1), the Bial Foundation (Grant 44/16), the Academy of Medical Sciences Springboard Award (SBF008\1113) and the Essex ESNEFT Psychological Research Unit for Behaviour, Health and Wellbeing (RCP15313) to Alejandra Sel and the Wellcome Trust to Matthew F. Rushworth (WT100973AIA). We also thank Miriam Klein-Flugge for helpful comments on the manuscript.

## Author contributions

E.F., M.R., A.S. designed the experiment; Y.C., B.P., A.S. collected the data; E.F., B.H., A.S. analysed the data; E.F., M.R. and A.S. wrote the manuscript. All authors discussed the results and implications and commented on the manuscript at all stages.

## Competing interests

The authors declare no competing interests.
