## [Peer Review File · Nature Communications]

REVIEWER COMMENTS

Reviewer #1 (Remarks to the Author):

This paper describes the results of a study designed to test the hypothesis that prediction errors in reward learning tasks are modulated by the phase of the cardiac cycle. This is an extension of a well established finding that perceptual sensitivity is greatest when exteroceptive stimuli are presented in diastole compared with systole. The authors show a modulation in CRS amplitude as a function of PE as well as a modulation in this CRS as a function of the phase of the cardiac cycle. I thought that this was an interesting, well conducted and well written study. I have a couple of questions.

1) I am not familiar with the tasks and models used in this study. Are the PEs a function of the stimulus presentation or do they in anyway depend on the participants responses? My apologies for not understanding this when reading the paper, but if it the latter I was interested in how varied these values are between participants. Either way it would be good to have some results describing the PE values and variations across task, participants or both (if applicable).

2) When fitting the models to the data is there anyway to report how well the model fitted and not just which model fitted better? Is the model a good model of the behaviour of the subject?

3) Is there anyway in which PE and arousal are linked here? Is PE arousing in the sense that the cardiac dynamics change. For example is the IBI interval modulated by the magnitude of the PE - decreasing with higher PE - this would be easy to show with the existing data and would be of interest here

Reviewer #2 (Remarks to the Author):

The present study investigates the relationship between cardiac activity and participants' sensitivity to outcomes in a reinforcement learning task. Healthy humans perform the learning task while EEG and EGG are recorded. Outcomes are presented for four seconds to allow for up to four heartbeats to occur during this period.

The key finding is that centro-parietal heartbeat-locked EEG responses during outcomes differs between trials with low vs high surprise (unsigned prediction error, PE), a result that is also obtained with multivariate analyses. This difference is present exclusively during the first heartbeat during the outcome presentation. Furthermore, heartbeat-locked EEG responses to outcomes differ as a function of cardiac phase, that is, depending on whether the outcome occurred during systole or diastole. Across subjects, the extent to which outcomes had a stronger (negative) effect on EEG during diastole vs systole correlated with learning rates.

The manuscript presents some interesting findings but I also have a number of concerns and questions.

1) In the introduction, the authors outline the two key quantities of interest in reinforcement learning (RL), lines 61-65: "Within this framework, learning is driven by two separate outcome dimensions: the signed PE, representing how much better or worse the value of an outcome is compared to what was

expected, and the absolute PE (also called 'saliency' or 'surprise'), representing how much an outcome differs from expectations regardless of whether it is better or worse."

I would disagree here. The key quantity adjusting the value estimate is the signed PE, at least in the algorithms I'm aware of. In contrast, the unsigned PE cannot be used to update values (as, by definition, it ignores the direction of surprise). Later, in the discussion (lines 380-381), it then says "... absolute PE, also called saliency impacts future attentional engagement...". This statement is more what I would agree with.

That said, the unsigned PE does however feature in algorithms like Pearce-Hall, where the (history of) outcome surprise is used to modulate learning rates (not to update value estimates, see above). Which brings me to my next point.

2) The major part of the manuscript revolves around cardiac-EEG relationships - heartbeat-locked EEG responses and (signed/unsigned PE). In contrast, only few links are made between cardiac activity and behaviour. Unless I'm missing something, the only link of that kind is presented in figure 4, where it is shown (across subjects) that participants with higher learning rates (and more points earned) are also those whose EEG response to outcomes differs more between diastole vs systole. This is an interesting finding, but I think it would be a lot more compelling to show this at the trial-wise level. To do so, one would need an estimate of how much PE are used to update values on each trial. There are RL models that feature a trial-specific learning rate, one being the above-mentioned Pearce-Hall associability. This would also have - in my view the advantage - that trial-specific learning rates are governed by the key quantity of interest to the authors - the recent history of unsigned prediction errors. If the authors could show e.g. that trial-specific learning rates differ between trials presented during systole vs diastole, I think this would provide strong support for their claim.

3) I also have some questions regarding the RL modelling.

a) It seems that only two models were tested, (i) a simple Rescorla-Wagner model with three free parameters (learning rate, inverse temperature, choice stickiness) and (ii) one that incorporated an additional weighting parameter that, at decision, allows for differential weighting of face vs house stimuli. One can easily think of further models. For instance, it appears highly plausible that, rather than giving differential weight to faces/houses at decision time, there is recency weighting at the time of learning - such that the second of the two sequentially presented stimuli is updated more strongly by the PE compared to the first (sort of an eligibility trace). Furthermore, an alternative scenario is that people do not learn individual stimulus values but conjunctive values for F-F, F-H, H-F and H-H. And of course, one could test (see previous comment) a model with trial-wise learning rates. I can imagine the latter will provide a worse fit compared to a simple model, but I think this is justifiable if the goal is to obtain trial-specific learning rates.

b) I think it would be great to have some more details on the modelling. The values for the fitted parameters are presented in figure 1, but perhaps a table presenting the average (across subjects) values would be good? Furthermore, can you also provide the (negative) log likelihood estimates - this would allow to get a better idea of how well the models captured participants' choices, rather than just BIC scores.

c) Please present a parameter recovery for the fitted model parameters.

d) I have the suspicion that learning rates and (inverse) temperature are highly correlated, owing to the (at least in some blocks) deterministic relationship between stimuli and outcomes. Please present the correlations between model parameters. Learning rates are high, in a number of cases = 1, which suggests single-shot learning. Which brings me back to my above point re alternative models, maybe even a simple (stochastic) win-stay/lose-switch model can provide a good account of participants' behaviour?

e) I am not sure about this, but I'm wondering to what extent it is a sensible choice to fit the models separately for the four different block types (thus, in each case to 2×60 trials). I wonder if this might result in overfitting. Did you expect learning rates (and the other parameters) to differ between block types? If not, I'd think the fitting of your parameters would be more robust if you applied it to the entire set of trials.

f) Model fitting procedure: Did you set upper/lower bounds for the parameter estimates? `fmincon`: you write (lines 562-564): "In addition, the estimated values of fitted parameters and negative log likelihoods were stable across the fitting procedure given different initial values." How do you know? I assume that `fmincon` was run multiple times from random starting points, but this is not detailed, at least I could not find this information.

4) The authors present results on the EEG response (heartbeat-evoked potentials) to signed (positive and negative) prediction errors. I would assume that positive and negative PE are highly correlated with outcome valence (reward/non-reward), so likely this presents a main effect of outcome? I would assume that this issue only pertains to signed, not to unsigned PE, but please report the correlations for both. If my assumption about correlation with signed PE is true, then these effects (figure 2B) should be interpreted as (most likely) reflecting outcome valence.

5) In figure 3, the authors use regularized Fisher discriminant analysis ("machine learning") on the HEP across all electrodes to distinguish high vs low surprise trials (highest vs lowest quantile of absolute PE). To me, this recapitulates the mass-univariate results from figure 2 with a multivariate approach - which likely confers higher sensitivity. However, the difference between positive and negative PE which was found in figure 2B is not observed here. Do you have an explanation for that?

An interesting finding in figure 3 is that high vs low surprise trials only differed locked to the 1st, but not 2nd or 3rd heartbeat during the outcome period. My understanding is that for the heartbeat-evoked potentials presented in figure 2, all heartbeats during the outcome period were used. If so, then the findings from figure 3D suggest to re-run this analysis including only responses locked to the 1st heartbeat. While post hoc, this may provide useful additional information.

6) Figure 4: I have difficulty understanding this figure, it seems that what is presented does not fully match the description in the text or legend. My understanding is that, here, responses to locked to the first heartbeat after outcome onset are split depending on whether the outcome occurred at systole vs diastole. Then it says (lines 303-306):

"Having split the outcomes [...], we found that the associated absolute PE-related CRS depended on whether the outcome was presented at systole or diastole, with the mean CRS magnitude related to absolute PE being more negative when outcomes were presented at the diastole compared to the systole phase ($t_{31}=2.8460$, $p=0.0078$, fig4b)."

I do not see where the absolute PE plays in here? To me, this indicates that heartbeat-locked responses during the outcome period are more negative during diastole vs systole, independent of outcome valence or surprise or anything else?

Jumping to figure 4E, this seems to speak to the above conclusion, however, I am again not sure whether this indeed provides evidence for a modulation of PE-responses by cardiac cycle: the authors describe a main effect of cardiac cycle and a main effect of signed PE - but no interaction. Thus, to me this describes that heartbeat-locked EEG outcome effects are (i) sensitive to high vs low surprise (as shown in figures 2/3) and (ii) are more pronounced during systole vs diastole (as shown in figure 4B), but not that PE-related responses are modulated?

Please also add error bars to 4E.

What does figure 4C show, is it the same as 4E, but only for signed PE near zero?

Figure 4D does not seem to be referred to in the text.

Figure 4F-K, the legend does not describe very well what is shown in the plots.

Minor:

1) The literature is not cited carefully. I find it alarming that of the six references I checked, none was actually a study that related to the findings cited in the text:

lines 430-432: "Neuronal excitability is influenced by the cardiac cycle; whilst neural signals from the baroreceptors occurring at systole attenuate concurrent brain activity (12, 13), and impair information processing, enhanced excitability and perceptual processing is observed during diastole (14, 15)" <-- Ref 12 is the first authors own work, 13 Leong et al., none of the two studies have reported effects related to the cardiac cycle. 14 is Hayden's ACC foraging study, 15 one of Schultz' many dopamine reviews, again (to my knowledge) nothing on cardiac cycle and neural excitability.

Likewise, (line 447) ref 19 (Kahnt & Tobler) does not measure "cardiac interoceptive sensitivity" and (lines 450-453) ref 20 (Behrens et al. 2007) is not at all about "Increased and decreased cardiac sensitivity has also been shown to help or hinder adaptive intuitive decision making..."

Please carefully check all references.

2) I would recommend not to refer to near-zero (absolute) PE as "near-threshold" - it may be misleading, and for some readers it might imply something about the stimulus being near perceptual threshold.

3) There is little behaviour in the results, primarily modelling. I think supplementary figure 1C shows learning curves, which I'd recommend to include in the main results. Looking at this curve, there seems to be surprisingly variable and near-chance performance in many blocks, barring a few exceptions.

4) Fig. 4E, y-axis label, what is STV?

Reviewer #3 (Remarks to the Author):

I read this report closely, and I have a mixed opinion of the conclusions. The authors demonstrate that prediction error learning precision is enhanced (i.e. better learning to low-information feedback) when feedback is timed to diastole in the cardiac cycle. Much of my hesitation is due to one major concern that needs to be addressed: there are a lot of reasons to be skeptical of heartbeat-evoked EEG activities. There were some other rather minor issues that could be addressed to help diminish the complexity of this report.

Major Concerns

1) The cardiac-related neural signal (CRS) or heartbeat evoked potential (HEP) is a controversial method. The authors note the methodological difficulty of linking the CRS since the cardiac field artifact (CFA) can be averaged into this potential. My concern is not with this electrical contaminant and whether or not it was removed via ICA. My concern is with the bolus-related movement of tissue or sensor, which is a possible separate artifact that is often overlooked in this field. I have followed a lot of work in this field, and I have never found this issue adequately addressed. I think there is a high probability that the slow peak and decline $\sim 100\text{ms} - 600\text{ms}$ following the R-spike is due to stereotyped movement of tissue or sensor with the pulsed blood flow, which becomes averaged together into a voltage change. For analogy, you could create a movement-related potential if you had a sensor on a grapefruit and you reliably tapped it in synchrony with a timing trigger. I think that addressing this potential confound is a prerequisite to interpreting a CRS.

2) The authors do show in Figure 4f-h that learning depends on CRS features, and only in learnable blocks. Yet this still could be an artifact of known effects of cardiac period on information intake (see minor point #2): the cardiac phase could influence learning and it could influence EEG, but EEG findings could be mediating or they could be spurious (i.e. major point #1 above). If the authors could demonstrate that this behavioral effect is intimately related to the findings shown from the CRS (e.g. mediating not spurious), then that would be very helpful for addressing this concern.

Minor Concerns

1) I spent a lot of time double checking what condition the absolute PE was derived from. I initially (correctly) assumed it was the absolute value of the feedback (+ or -) PEs. Yet Fig 1h led me to consider that the second stimulus presentation could be framed as an absolute PE since after the first stimulus it is unknown what the outcome will be, but the second stimulus will potentially resolve this uncertainty (with a salience PE). Thus I spent time understanding if the data were locked to that event, not the feedback. Figure 3a and 3e were helpful to resolve this confusion, but I think an alteration of 1h would be helpful. It would be helpful to have a very clear sentence in the text that formally defines absolute PE (line 240 helps, but it's a little bit of info, a little late).

2) There is old literature in psychophysiology on cardiac slowing during "information intake". The work from the Laceys and Obrist comes to mind, as well as some critical re-appraisals (see below). If the authors are interested, this would be a very interesting old corpus to integrate with this new approach. (<https://www.sciencedirect.com/science/article/abs/pii/0301051178900595>, <https://www.sciencedirect.com/science/article/abs/pii/0301051180900277>, <https://psycnet.apa.org/record/1973-10545-001?doi=1>, <https://onlinelibrary.wiley.com/doi/abs/10.1111/j.1469-8986.1970.tb02257.x>)

Reviewer #4 (Remarks to the Author):

I would like to thank the editor for inviting me to review this interesting work. I congratulate the authors for the originality and methodological rigor.

Still, it is not straightforward how to place these new findings among the existing literature. I recommend a revision on the results interpretation and a contextualization with respect to recent literature. My specific points below:

1. In the introduction's 1st and 3rd paragraphs seems that the authors agree on a binary view of perception, depending on the cardiac cycle phases. However, abundant evidence exists to the date showing that these heart-brain interactions are complex and non-linear phenomena. As described for instance in Silvani et al (10.1098/rsta.2015.0181) and Candia-Rivera et al (10.1016/j.crneur.2022.100050). Furthermore, in Skora et al (10.1016/j.neubiorev.2022.104655) is reviewed that this cardiac phase hypothesis does not go in the same direction in all reported studies. Indeed, in the results (Figure 4) what is shown is a trend.

2. How these results stand with the literature on decision and heart-brain interactions? The authors limit their comments in the discussion to general literature on interoception, and with respect to Galvez-Pol 2020, 2022 studies. I believe the discussion could be enriched in this regard.

Here are some studies showing correlates between heart-brain interactions and different dimensions of decision-making:

preference consistency (Azzalini et al, 10.1523/JNEUROSCI.1932-20.2021),

response to stimuli (Larra et al, 10.1038/s41598-020-61068-1),

action execution (Palser et al, 10.1016/j.cognition.2021.104907),

response inhibition (Rae et al, 10.1038/s41598-018-27513-y; Ren et al, 10.1016/j.biopsycho.2022.108323),

choice-reward (Fujimoto et al, 10.1073/pnas.2014781118).

3. Did the ICA computation include the ECG channel? If yes, how many leads? In the HEP literature exists a high heterogeneity on this preprocessing step, and therefore, it needs to be clarified. To exemplify, in some studies like in Couto et al (10.1016/j.autneu.2015.06.006) the CFA is not removed, in others like in Al et al (10.1073/pnas.1915629117) the ECG is not included in the ICA computation and CFA were removed only when found. In our recent guideline paper (10.1016/j.jneumeth.2021.109269) we recommended to include the ECG on the ICA computation, in addition to a final EEG average reference, as the authors apparently did.

4. The authors report on the methods, ICA part: “Those independent components (4.78 on average across participants; 1.13 SD)”. Is this the component number as sorted in Fieldtrip? Or is it the number of components?

5. Did the authors remove the CFA in all participants? If not, please report the number of participants in which the CFA was not found.

6. Is there a reason to use the acronym CRS? Why not just HEP or HEP with a subscript?

7. Could you clarify on the methods how many epochs (or trials) are being averaged in each CRS? This could be relevant if only one heartbeat is considered after the outcomes. If it is only one epoch, it needs to be clearly justified and discussed whether it is a reliable measurement.

You will find below detailed responses to the reviewers' comments. Reviewers' comments are numbered and in normal font. Responses are in blue, extracts from the original manuscript are in italic and black while *the revised portions are in red*.

Reviewer #1:

This paper describes the results of a study designed to test the hypothesis that prediction errors in reward learning tasks are modulated by the phase of the cardiac cycle. This is an extension of a well-established finding that perceptual sensitivity is greatest when exteroceptive stimuli are presented in diastole compared with systole. The authors show a modulation in CRS amplitude as a function of PE as well as a modulation in this CRS as a function of the phase of the cardiac cycle. I thought that this was an interesting, well conducted and well written study. I have a couple of questions.

We thank the reviewer for highlighting that our study is interesting, well conducted and well written. We have now amended the manuscript to better highlight how the model fits the behaviours and performed additional analysis to provide more supporting evidence for our main conclusions.

1) I am not familiar with the tasks and models used in this study. Are the PEs a function of the stimulus presentation or do they in any way depend on the participants' responses? My apologies for not understanding this when reading the paper, but if it is the latter I was interested in how varied these values are between participants. Either way it would be good to have some results describing the PE values and variations across tasks, participants or both (if applicable).

Many thanks for raising this point. In reinforcement learning models, PEs are a function of the stimulus presentation and the participants' responses. Although the stimuli presentations were controlled, variation in individual performance and speed of learning may result in fluctuations of PEs across the blocks. In our model, the parameter controlling the speed of learning is the learning rate. The learning rate scales the learning rule; it determines the extent to which the PE changes – or updates – options' values. PEs are represented in Figure 1H as schematic and hypothesised profiles. Each coloured pattern illustrates a different way in which activity in a neural structure may manifest if it is sensitive to different aspects of outcomes and their associated prediction error (PE). The blue pattern illustrates how activity will appear if it simply reflects the valence of an outcome – correct or incorrect – in a categorical manner as either positive or negative; responses are positive whenever the outcome is correct and negative whenever the choice is incorrect. The red pattern shows a monotonically increasing response profile consistent with a continually varying PE representation where activity does not just reflect whether the outcome is better or worse than expected but by how much that is the case. The black pattern, like the red pattern, is continually varying, as a function of the difference between the outcome and the expectation but now information about the sign of the difference, whether it is better or worse than expected, has been lost. It is therefore often referred to as an unsigned PE. It captures surprise effects with greater responses when the outcome deviates further from expectation in either direction, independent of the sign (valence) of the PE.

We also agree with the reviewer that, as the HEP is constructed along different dimensions of PE values, it is important to illustrate how these PEs vary across blocks. We have now added two panels to do this in the main first figure shown below (panels I-L). Corrections of the original manuscript are presented in red.

Figure 1. Schematic representation of the task and RL results for all four association schemes, highly predictive anticorrelated (HA), highly predictive correlated (HC), variable predictive (VP) and non-predictive (NP). **(a)** task + cardiac-related neural signals (stHEP) recorded at outcome for 4 sec - enough to get on average 3.5 HEPs **(b)** example of association between cues and predicted colour for one version of the task (anti-correlated blocks – see Methods) **(c)** prediction strengths for each block **(d)** model prediction of choices **(e)** Learning rates from the best fitting model **(f-g)** Reaction times and performance (mean reward across the four blocks) **(h Left panel)** Definition of absolute and signed PE. **(h right panel)** Each coloured pattern illustrates a different way in which activity in a neural structure may manifest if it is sensitive to different aspects of outcomes and their associated PE (locked at time of feedback). The blue pattern illustrates activity as a function of outcome valence – in a categorical manner as either positive or negative. The red pattern shows a monotonically increasing response profile consistent with a continually varying PE representation. The black pattern is continually varying as a function of the difference between the outcome and the expectation in an unsigned fashion. **(i)** Signed PEs are presented across participants for all blocks, following the example of a task like in **(c)**. **(j)** Variation of PEs across blocks **(k-l)** similar to **(i-j)** for absolute PEs.

2) When fitting the models to the data is there any way to report how well the model fitted and not just which model fitted better? Is the model a good model of the behaviour of the subject?

This is a good point. We are sorry this was not obvious in the previous version of our manuscript, although it was presented on supplementary figure 1. We have now added information about the model fit and in response to another reviewer, tested additional models as well. For all models, we estimated free parameters by likelihood maximisation and Laplace approximation of model evidence to calculate the Bayesian Information Criterion (BIC) and the exceedance probability respectively (this can now be found in the new supplementary figure 1). Bayesian model comparison revealed that the SimC model, that we originally proposed, explains the data better than other more complex models (Lower BIC values indicate better model fit, SimC model: BIC=9350, RW model: BIC=9370, DYNA model: BIC=9400, CUNJ: BIC=9390). Exceedance probabilities for the models based on approximate posterior probabilities suggested that our SimC model outperformed the others ($p=0.95$; supplementary figure 1C insert).

To confirm the relationship between the better model's predictions and behaviour, we compared choice probabilities predicted by the SimC model and the actual recorded frequencies of people's actual choices and found that the model matched behaviour well (supplementary figure 1). A further linear regression analysis at the individual level generated an average R^2 of 0.94 across people and regression coefficients that were significant ($P<0.001$) for each person. This means that, on average, most of the trial-to-trial variation in participants' choices is well explained by the model. So, to answer the reviewer's question – is it a good model of the behaviour of the subject – yes, it is a good model. Having established the goodness of fit of the SimC model to behaviour, all further analyses were conducted using the PEs estimated with this model. We have presented below in supplementary figure 1D, the model-predicted choice probabilities (y-axis) derived from the SimC algorithm (binned into four bins – bin size of 0.25 - and averaged across all subjects and across symbols) closely matched participants observed behavioural choices (x-axis), calculated for each bin as the fraction of trials in which they chose one colour.

Supplementary Figure 1. (A) Summed log-model evidence using a variational inversion scheme for all models. Higher scores indicate better fit. (B) Nested model comparison using the summed integrated Bayesian Information Criterion (BICint) scores for all models. Lower scores indicate better fit. RL_HEP_SimC is the winning model. (C) Comparisons of Bayes factors between best and second-best models. Bars are individual blocks sessions and direction of the bars (left vs right) indicate better model fit in favour of the RL_HEP_SimC (going to the right) compared to RL_HEP_rw (going to the left). (Small panel) Exceedance probability also favours the SimC model among our set of candidate models. The dashed red line indicates an exceedance probability of 0.95. (D) Model-predicted choice probabilities (y-axis) derived from the SimC algorithm (binned into four bins – bin size of 0.25 - and averaged across all subjects and across symbols) closely matched participants observed behavioural choices (x-axis), calculated for each bin as the fraction of trials in which they chose one colour.

3) Is there any way in which PE and arousal are linked here? Is PE arousing in the sense that the cardiac dynamics change. For example is the IBI interval modulated by the magnitude of the PE - decreasing with higher PE - this would be easy to show with the existing data and would be of interest here

We thank the reviewer for this excellent suggestion. To implement it, we calculated the mean IBI on each trial (across the multiple heart beats collected at time of outcome) to test whether this value could predict PE or the salience/ magnitude of the PE ($|PE|$ or unsigned PE). We did not find any relationship between the two: PE: $p = 0.4567$, conf int = $[-0.0130 \quad 0.0060]$, tstat:

-0.7537, df: 31, sd: 0.0264, and abs(PE): $p = 0.2223$, $c = [-0.0277 \quad 0.0067]$, $tstat: -1.2455$, df: 31, sd: 0.0477. Note that it is possible that the arousal response coded by the absPE does not necessarily need to translate into large physiological changes such as changes in heartbeat or skin conductivity, although some studies have reported changes in pupil dilation in response to changes in surprise (Preuschoff 2011, *Frontiers in Neuroscience*; Aston-Jones & Cohen, 2005, *Annual Review of Neuroscience*).

Reviewer #2:

The present study investigates the relationship between cardiac activity and participants' sensitivity to outcomes in a reinforcement learning task. Healthy humans perform the learning task while EEG and EGG are recorded. Outcomes are presented for four seconds to allow for up to four heartbeats to occur during this period. The key finding is that centro-parietal heartbeat-locked EEG responses during outcomes differ between trials with low vs high surprise (unsigned prediction error, PE), a result that is also obtained with multivariate analyses. This difference is present exclusively during the first heartbeat during the outcome presentation. Furthermore, heartbeat-locked EEG responses to outcomes differ as a function of cardiac phase, that is, depending on whether the outcome occurred during systole or diastole. Across subjects, the extent to which outcomes had a stronger (negative) effect on EEG during diastole vs systole correlated with learning rates. The manuscript presents some interesting findings but I also have a number of concerns and questions.

We thank the reviewer for highlighting that our study is interesting. Based on the reviewer's comments (see below for details) we have now amended the manuscript to better highlight the novelty of this work. As a result of the reviewer's interesting suggestions, we have now performed several additional analyses to provide more supporting evidence for our main conclusions.

1) In the introduction, the authors outline the two key quantities of interest in reinforcement learning (RL), lines 61-65: "Within this framework, learning is driven by two separate outcome dimensions: the signed PE, representing how much better or worse the value of an outcome is compared to what was expected, and the absolute PE (also called 'saliency' or 'surprise'), representing how much an outcome differs from expectations regardless of whether it is better or worse." I would disagree here. The key quantity adjusting the value estimate is the signed PE, at least in the algorithms I'm aware of. In contrast, the unsigned PE cannot be used to update values (as, by definition, it ignores the direction of surprise). Later, in the discussion (lines 380-381), it then says "... absolute PE, also called saliency, impacts future attentional engagement...". This statement is more what I would agree with. That said, the unsigned PE does however feature in algorithms like Pearce-Hall, where the (history of) outcome surprise is used to modulate learning rates (not to update value estimates, see above). Which brings me to my next point.

We agree with the reviewer that it is important to ensure that the terminology is used precisely. We now emphasise that typical RL models - such as the one used in the study - does not explicitly use the absolute PE to update value estimates, although some other models do (see methods section). We make clear in the revised manuscript that if this happens it is because the impact that each PE has on the updating of a value estimate can vary and that this variation is indexed by a parameter often called the learning rate. We then note that absolute PE may, in turn, influence the learning rate, as in the Pearce-Hall model. However, we emphasise that the two quantities, PE and absolute PE, are associated with variation in largely distinct distributed neural representations and thus carry separate roles. As the

reviewers mentions, one of these roles is to drive the organism towards allocating more attention and, as a further consequence, this may lead to greater future adjustments in the organism's estimates of choice values. We amended the introduction as follow and we modified the references to correctly reflect this point:

Adaptive decisions rely on accurate subjective value estimates associated with past experience of choices and their consequences. These values can be formally defined through the reinforcement learning framework (Sutton & Barto, 2018) that uses the difference between expectation and outcome (the PE) to update values associated with choices. A choice that led to a positive outcome is more likely to be associated with a higher value than a choice that did not. While the signed PE represents how much better or worse the value of an outcome is compared to what was expected, the absolute PE (also called 'salience', 'surprise', or unsigned PE) represents how much an outcome differs from expectations regardless of whether it is better or worse (Pearce & Hall, 1980). Activity in separate neural networks has been related to the signed PE and absolute PE (Fouragnan et al., 2017, 2018; Queirazza et al., 2019). It has thus been hypothesised that these two quantities represent two different dimensions of learning. Whereas positive and negative signed PEs lead to the reinforcement or extinction of the choices that led to them (Dayan & Daw, 2008), the absolute PEs can determine the extent to which the associations between outcome and expectations need to be adjusted (Fouragnan et al., 2017; Rouhani & Niv, 2021).

2) The major part of the manuscript revolves around cardiac-EEG relationships - heartbeat-locked EEG responses and and (signed/unsigned PE). In contrast, only a few links are made between cardiac activity and behaviour. Unless I'm missing something, the only link of that kind is presented in figure 4, where it is shown (across subjects) that participants with higher learning rates (and more points earned) are also those whose EEG response to outcomes differs more between diastole vs systole. This is an interesting finding, but I think it would be a lot more compelling to show this at the trial-wise level. To do so, one would need an estimate of how much PE is used to update values on each trial. There are RL models that feature a trial-specific learning rate, one being the above-mentioned Pearce-Hall associability. This would also have - in my view the advantage - that trial-specific learning rates are governed by the key quantity of interest to the authors - the recent history of unsigned prediction errors. If the authors could show e.g. that trial-specific learning rates differ between trials presented during systole vs diastole, I think this would provide strong support for their claim.

We thank the reviewer for suggesting this interesting new analysis. We agree that the trial-wise relationship between the HEP fluctuation and model prediction can be exploited further to strengthen our results. We have decided to investigate **two new avenues**.

In the first, as suggested by the reviewer, we have developed multiple new RL models including one model incorporating the characteristics that the reviewer highlights: a model in which the learning rate scales with the slope of the smoothed $|PE|$. This model reprises

Pearce-Hall's theory that surprise drives the acquisition of stochastic stimulus-outcome contingencies. In this new model, the smoothing of the unsigned $|PE|$ (the degree of which is regulated by a free parameter ρ) should render the inference process about whether a change has occurred in the environment more robust to any inherent task stochasticity. Moreover, an additional free parameter γ controls the extent to which the dynamic updating of the learning rate is influenced by the slope. For example, whilst lower values of γ yield substantial trial-by-trial changes of the dynamic learning rate even in the presence of small slope estimates (that is, low surprise), higher values of γ result in a more stable learning rate even in the presence of significant slope estimates (that is, high surprise). Hence, this model also allows for the possibility that subjects might be employing a relatively fixed learning rate.

Caption. Representation of the dynamic learning rates over time, across the learning blocks (every 60 trials).

With this model, we tested whether dynamic learning rates were different across the two phases of the cardiac cycle. This analysis did not show any difference between the systole and diastole phases. It is important to note that it was already the case that neither trial by trial variation of PE or $|PE|$ showed differential mean values between the two cardiac cycles either. We had presented this in the paper but realised now this was perhaps not clear enough. We are now very careful that this is better emphasised in the paper.

Naturally, as the diastole phase is longer on average than the systole phase, we expect a higher number of outcomes presented during the diastole phase ($m=65\pm5$ and $54\pm5 - N: 32$, mean and SD for diastole and systole, respectively). We also found that all other aspects of the task were identical in these conditions. For example, and importantly, the frequency of occurrence of both conditions was similar in the different learning blocks (e.g. predictive or non-predictive) employed in the task (see Methods; systole $F_3=0.2$, $p=0.893$, $\eta^2=0.0049$; diastole $F_3=0.2$, $p=0.896$, $\eta^2=0.0048$, supplementary Fig.3a). The two phases were also associated with similar levels of overall reward received in the task ($t_{31}=0.8046$, $p=0.4272$,

CI=[-0.008 0.02], Cohen D=0.14, supplementary Fig.3b), unsigned PEs from the RL model ($t_{31}=-0.0581$, $p=0.9541$, CI=[-0.026 0.025], Cohen D=-0.01, Fig.4d) or signed PE ($t_{31}=-0.72$, $p=0.4769$, CI=[-0.042 0.02], Cohen D=-0.12). This is important as it shows that the cardiac cycle does not modulate model estimates in a direct way.

*Having split the outcomes according to the cardiac cycle, we then moved on to test whether the associated **absPE-HEP** depended on whether the outcome was presented at systole or diastole. This is indeed what we found. The mean **absPE-HEP** was more negative when outcomes were presented at the diastole compared to the systole phase (t-test: $t_{31}=2.8460$, $p=0.0078$, CI=[0.0107 0.065], Cohen D=0.55, fig4b). Similarly, we tested, with a mixed effects linear model, whether the STV could be predicted by a more complex model, including the trial-by-trial model-based absolute PE which is expected to covary with STV, the heart cycle (categorical variable) and the interaction term. Beyond the linear relationship between absolute PE and single-trial HEP which is to be expected (also see fig3c) - such that the effect of the cardiac cycle increased as absolute PEs became smaller (main effect of absPE in mixed effect model: $t_{122}=3.539$, $p=0.0005$, 95% CI, [0.11 0.39], Partial $\text{Eta}^2 = 0.644$), we also found a main effect of cardiac cycle on the STV (main effect of heart cycle in mixed effect model: $t_{122}=-2.336$, $p=0.021$, 95% CI, [-0.35 -0.03], Partial $\text{Eta}^2 = 0.0497$). This effect was driven by the fact that near-threshold absolute PE were more strongly represented at diastole (interaction effect: heart cycle*absolute PE: $t_{122}=1.96$, $p=0.052$, 95% CI, [-0.0008 0.2], Partial $\text{Eta}^2 = 0.0345$). We illustrate this result with another analysis, independent of the previous one, showing the difference between the STV **absPE-HEP** < 50% percentile for systole and diastole (Fig 4c). This confirms that timing within the cardiac cycle modulates neural signals of absolute PE and that these representations are stronger after an outcome is presented at diastole compared to systole.*

We argue that this is consistent with what we had previously reported; it is not the model estimates that are different between the two phases of the cardiac cycle but the cardiac related trial by trial neural signals reflecting absolute PE, which we refer to as **absPE-HEP** for clarity.

Secondly however, and in spirit of the reviewer's suggestion that it would be wise to pay more attention to the varying impact of information provided to participants on single trials as function of cardiac cycle, we believe we have come up with a way to test the idea at the heart of the reviewer's comment; that it should be possible to see a relationship between trial-by-trial variation in **absPE-HEP** and trial by trial variation in updating in the values estimated for each choice. To do this we investigated whether the **absPE-HEP** at outcome (t) could predict change in choices in the next trial at (t+1). To do so, we ran an additional regression to predict choice switching behaviours at (t+1) with the residual **absPE-HEP** variance at (t) after controlling for the part of the **absPE-HEP** that was collinear with the absolute PE and added the systole and diastole as a separate regressor. Our results showed that participants were more likely to switch after a highly surprising outcome (absPE: $p = 1.4791e-13$, tstat: 12.4041, df: 31) aligning with previous reports (Fouragnan et al. 2017) but also that higher residual variance in **absPE-HEP**, even after controlling for that part of the HEP that was collinear with

absolute prediction error, also predicted switch behaviours on the next trials ($absPE_HEP$: $P = 0.0162$, $tstat: -2.5440$, $df: 31$). In addition to this key result, the analysis, also revealed that there was a trend towards significance suggesting that the cardiac cycle on the previous trials (whether the outcome's onset happened at systole or diastole) also predicted a switch (i.e. whether the outcome onset occurred at diastole or systole on trials prior to t tended to predict choice switching on trial $t+1$: $SysDias$: $p = 0.0591$, $tstat: -1.9592$, $df: 31$). Interestingly, this seems to align with our previous results showing that *absPE-HEP* at outcome are stronger at diastole than systole. In a similar way, the outcomes falling at diastole tend to predict switch behaviours more strongly than outcomes falling at systole. There were no significant interaction effects in this analysis. By contrast other aspects of behaviour unrelated to learning such as reaction times RTs) could not be predicted except by $absPE$: $p = 0.0120$, $tstat: 2.6684$, $df: 31$.

Caption. Left: A logistic regression analysis showed that switch / stay (1 / 0) could be predicted by several predictors, including the Absolute PEs (participants were more likely to switch after a highly surprising outcome) but also residual cardiac related signals. This figure is now presented on Figure 4 of the main manuscript. **Right:** This was not the case for a linear regression model predicting Response Times.

We are now presenting this in the text and in the new Figure 4, panel e:

Figure 4. Influence of the cardiac cycle on the *absPE-HEP* and learning. **(a)** Schematic description of the systole and diastole phases. In red and blue are the systole and diastole periods respectively. Below is a representation of two example trials on which outcome onsets happened at systole or diastole. We then looked at the EEG response locked to the next heart-beat. **(b)** Mean amplitude difference between the *absPE-HEP* of the first heartbeat after all outcomes presented at systole versus diastole ($N=32$). A violin plot is used to present all individual participants' averages. **(c)** Mean amplitude difference in the *absPE-HEP* for the low salient outcomes presented at systole versus diastole across subjects ($N=32$). **(d)** Mean amplitude difference in *absPE-HEP* for the high salient outcomes presented at systole versus diastole ($N=32$). **(e)** A logistic regression analysis showed that switch / stay (1 / 0) could be predicted by several predictors, including the Absolute PEs (participants were more likely to switch after a highly surprising outcome) but also residual *absPE-HEP*. **(f-k)** Results of the correlation between the regression coefficient for each participant between *absPE-HEP* and systole/distole and the mean reward and learning rates in the task. In red, the fit of the robust regression. Any of these results remain true even when including a covariate indexing features of the external outcome type – reward and absolute PE from the model. Particularly, in **(f)** learning rates – all task blocks **(g)** learning rates – predictive blocks **(h)** learning rates – non predictive blocks **(i)** reward – all task blocks **(j)** reward - predictive blocks **(k)** reward – non predictive blocks.

3) I also have some questions regarding the RL modelling.

a) It seems that only two models were tested, (i) a simple Rescorla-Wagner model with three free parameters (learning rate, inverse temperature, choice stickiness) and (ii) one that incorporated an additional weighting parameter that, at decision, allows for differential weighting of face vs house stimuli. One can easily think of further models. For instance, it appears highly plausible that, rather than giving differential weight to faces/houses at

decision time, there is **recency weighting at the time of learning** - such that the second of the two sequentially presented stimuli is updated more strongly by the PE compared to the first (sort of an eligibility trace). Furthermore, an alternative scenario is that people do not learn individual stimulus values but **conjunctive values for F-F, F-H, H-F and H-H**. And of course, one could test (see previous comment) a model with **trial-wise learning rates**. I can imagine the latter will provide a worse fit compared to a simple model, but I think this is justifiable if the goal is to obtain trial-specific learning rates.

We thank the reviewer for the advice. We have now revised our manuscript and tested three additional models and their fit to behaviours in addition to our original model.

The first model learns separate values for Face and House, (RL_HEP_SimC) as described in the original paper. In the second model, the conjunction values are learnt separately, resulting in three separate pairs from which values are learnt (FF, FH, and HH) (RL_HEP_CONJ). Note that we also tried four pairs (with FH and HF) and the fit was worse. In these two first models, the Prediction Variable PV is computed either after summing up the equally weighted stimulus–outcome association strengths for each cue V1 and V2 that is updated after each cue is presented (RL_HEP_SimC) or after a Face-Face V1, Face-House V2 or House-House V3 (RL_HEP_CONJ) is presented. PE is the prediction error. PV is then converted to a choice probability following the equation: $p = 1/(1 + e(\beta * (PV - 0.5) + \gamma * C_{n-1}))$, where β is the inverse temperature, or exploration parameter and γ represents the choice stickiness (the degree to which choices are likely to simply be repeated from trial to trial regardless of outcome). C_{n-1} is the choice in the previous trial, O is the outcome in the current trial and α is the learning rate shared by both items. The subscript n represents the current trial, and n + 1 represents the updated trial. There are three free parameters in these two first models: the learning rate α , the exploration parameter β , and the choice stickiness factor γ .

The third model has trial-wise learning rates, which we refer to as the dynamic learning rate model (RL_HEP_DYNA), as described in the previous answer. In this model, an additional free parameter *gamma* controls the extent to which the dynamic updating of the learning rate is influenced by the slope.

The fourth model has a recency weighting at the time of learning which updated the value estimate for the second of the two presented cues more than the first (RL_HEP_RW). In the RW model, the value of the most recently presented item (the second), is more strongly updated than the first one as a function of an additional free parameter *trace*.

For all models, we estimated free parameters by likelihood maximisation and Laplace approximation of model evidence to calculate the Bayesian Information Criterion (BIC) and the exceedance probability respectively (this can now be found in the new supplementary figure 1). Bayesian model comparison revealed that the RL_HEP_SimC model that we originally proposed explains the data better than other more complex models (Lower BIC values indicate better model fit, RL_HEP_SimC model: BIC=9350, RL_HEP_RW model: BIC=9370, RL_HEP_DYNA model: BIC=9400, RL_HEP_CONJ model: BIC=9390). Exceedance

probabilities for the models based on approximate posterior probabilities suggested that our RL_HEP_SimC model outperformed the others ($\bar{r}=0.95$; supplementary figure 1C insert).

To confirm the relationship between the better model's predictions and behaviour, we compared choice probabilities predicted by the RL_HEP_SimC model and the actual recorded frequencies of people's actual choices and found that the model matched behaviour well (supplementary figure 1). A further linear regression analysis at the individual level generated an average R^2 of 0.94 across people and regression coefficients that were significant ($P < 0.001$) for each person. This means that, on average, most of the trial-to-trial variation in participants' choices is well explained by the model. Having established the goodness of fit of the RL_HEP_SimC model to behaviour, all further analyses were conducted using the PEs estimated with this model. We have presented below in supplementary figure 1D, the model-predicted choice probabilities (y-axis) derived from the RL_HEP_SimC algorithm (binned into four bins – bin size of 0.25 - and averaged across all subjects and across symbols) closely matched participants observed behavioural choices (x-axis), calculated for each bin as the fraction of trials in which they chose one colour.

To answer the reviewer's constructive comment, the summed integrated BIC (Huys et al, 2012) is now presented for all models in comparison with our RL_HEP_SimC model as well as the log-evidence. Although we have fitted these four models as well as their combinations, for simplicity and parsimony, we only present the four main models. Note that no combinatory composition fits the data better than their simpler counterpart.

Supplementary Figure 1. (A) Summed log-model evidence using a variational inversion scheme for all models. Higher scores indicate better fit. (B) Nested model comparison using

the summed integrated Bayesian Information Criterion (BICint) scores for all models. Lower scores indicate better fit. RL_HEP_SimC is the winning model. (C) Comparisons of Bayes factors between best and second-best models. Bars are individual blocks sessions and direction of the bars (left vs right) indicate better model fit in favour of the RL_HEP_SimC (going to the right) compared to RL_HEP_rw (going to the left). (Small panel) Exceedance probability also favours the SimC model among our set of candidate models. The dashed red line indicates an exceedance probability of 0.95. (D) Model-predicted choice probabilities (y-axis) derived from the SimC algorithm (binned into four bins – bin size of 0.25 - and averaged across all subjects and across symbols) closely matched participants observed behavioural choices (x-axis), calculated for each bin as the fraction of trials in which they chose one colour.

Importantly, we used the PEs and |PEs| to split our HEP to extract their single-trial information via a regularised Fisher discriminant (that ultimately split the data in a binary fashion, like a logistic regression). We thus tested whether the PEs computed across the different RL models differed compared to the RL_HEP_SimC. We found that all PEs strongly correlated with the PE values from the original model ($r = 0.9488$, $ste = 0.0056$) and as such the additional variance from the newly computed PEs when implementing our machine learning methods would be negligible. We also looked at the split between high/low PEs as well as high/low |PE| (splits which would then inform the multivariate analysis of the HEP data) and the overall overlap was >95% rendering a new analysis almost identical to the one we are presenting in the paper.

Caption. Overlap of all PEs across all models.

b) I think it would be great to have some more details on the modelling. The values for the fitted parameters are presented in figure 1, but perhaps a table presenting the average (across subjects) values would be good? Furthermore, can you also provide the (negative) log likelihood estimates - this would allow to get a better idea of how well the models captured participants' choices, rather than just BIC scores.

Thank you for this suggestion. Following the reviewer's advice, we have now added a supplementary table showing the average (across subjects) values for all parameters (supplementary table 1).

Parameter estimate for best behavioural model, depicted as mean \pm SE

	HC	HA	VP	NP
Learning rate	0.7965 \pm 0.1454	0.8274 \pm 0.1961	0.6912 \pm 0.1648	0.6036 \pm 0.1206
SoftMax inv. Temp	0.1992 \pm 0.2202	3.9277 \pm 2.2033	0.1190 \pm 0.2169	0.0388 \pm 0.1050
Choice stickiness	-0.2351 \pm 0.2945	0.7982 \pm 0.2808	0.1908 \pm 0.3010	-0.0546 \pm 0.312

Supplementary table 1. Averaged best-fitting parameter estimates (across subjects) \pm SE

This can be found in the supplementary material of the revised manuscript. Regarding the neg log likelihood, we have now presented this in supplementary figure 1 (see aforementioned response).

c) Please present a parameter recovery for the fitted model parameters.

This can be found in the supplementary table 2.

	HC	HA	VP	NP
Learning rate	0.7965 \pm 0.1454	0.8274 \pm 0.1961	0.6912 \pm 0.1648	0.6036 \pm 0.1206
	0.6535 \pm 0.1270	0.6208 \pm 0.1873	0.4870 \pm 0.1222	0.5831 \pm 0.1088
SoftMax inv. Temp	0.1992 \pm 0.2202	3.9277 \pm 2.2033	0.1190 \pm 0.2169	0.0388 \pm 0.1050
	2.1726 \pm 1.9195	7.3658 \pm 3.0089	1.2625 \pm 1.9670	0.2412 \pm 0.8831
Choice stickiness	-0.2351 \pm 0.2945	0.7982 \pm 0.2808	0.1908 \pm 0.3010	-0.0546 \pm 0.3120
	-0.7287 \pm 0.3292	-0.5099 \pm 0.3780	-0.3729 \pm 0.3496	-0.4951 \pm 0.3134

Supplementary table 2. Recovery parameters estimates (across subjects) \pm SE. Parameters are presented below the ones found during the fitting procedure.

d) I have the suspicion that learning rates and (inverse) temperature are highly correlated, owing to the (at least in some blocks) deterministic relationship between stimuli and outcomes. Please present the correlations between model parameters. Learning rates are high, in a number of case = 1, which suggests single-shot learning. Which brings me back to my above point re alternative models, maybe even a simple (stochastic) win-stay/lose-switch model can provide a good account of participants' behaviour?

Many thanks for raising this point. We have tested the relationship between the learning rates and (inverse) temperature. While the relationship is indeed positive, it was not strongly

correlated (0.2836 across all learning blocks and all parameters), mainly due to the variable and non-predictive blocks.

The WStLSw analysis is an excellent idea, which we have now tested (as you have seen a Switch/Stay approach was also used when responding to the reviewer’s point 2 above). In the revised manuscript we introduce this section by explaining, as the reviewer points out, that a model free approach is warranted because we observed that learning rates were often high. We thus thought that it would be interesting to run a similar model as the one presented previously, showing that Switch/Stay (SwSt: 1 / 0 respectively) could be predicted by several predictors, including the |PEs| (participants were more likely to switch after a highly surprising outcome) but also cardiac related signals. We thus ran a similar model but, in this case, trying to predict a WStLSw pattern (WSt=1, LSw=-1, 0 elsewhere).

The logistic models ran were:

$$1) WStLSw \sim SysDias + SingleTrialHEP + ABSPE + SingleTrial*SysDias*ABSPE$$

$$2) SwSt \sim SysDias + SingleTrialHEP + ABSPE + SingleTrial*SysDias*ABSPE$$

Because these models are similar in terms of their complexity, we compare their fits to the data. This shows that the SwSt (Switch/Stay model) provides a better account of the data than a WStLSw. The SwSt BIC is 20326. For the WStLSw, the BIC is 38820.

Caption. (e) A logistic regression analysis showed that win-stay/lose-switch and switch / stay could be predicted by the Absolute PEs (participants were more likely to switch after a highly surprising outcome) but also residual *absPE-HEP*.

e) I am not sure about this, but I'm wondering to what extent it is a sensible choice to fit the models separately for the four different block types (thus, in each case to 2*60 trials). I wonder if this might result in overfitting. Did you expect learning rates (and the other parameters) to differ between block types? If not, I'd think the fitting of your parameters would be more robust if you applied it to the entire set of trials.

This is an interesting point and one that we have spent some time considering too. In the end, however, we concluded that differences in degree of association are likely to lead to change

in the key parameters such as uncertainty and surprise that are widely thought to, in some way, influence learning rate (Pearce and Hall, 1980; Behrens et al., 2007; Fouragnan et al., 2017; 2018). We note that the use of some of the additional indices that we now employ, as a result of the reviewer's suggestions, such as win-stay-lose-switch analyses provide alternative less model-dependent analysis approaches.

f) Model fitting procedure: Did you set upper/lower bounds for the parameter estimates? fmincon: you write (lines 562-564): "In addition, the estimated values of fitted parameters and negative log likelihoods were stable across the fitting procedure given different initial values." How do you know? I assume that fmincon was run multiple times from random starting points, but this is not detailed, at least I could not find this information.

We have now changed our fitting procedure based on an iterative expectation-maximisation (EM) algorithm to fit the four models described earlier (Huys et al, 2012; Wittmann et al; 2019). We used an iterative expectation-maximisation (EM) algorithm to fit the models. During the expectation step, we calculated the maximum posterior likelihood estimate obtained with the parameter vector \mathbf{h}_i of each session i given the observed choices and given group-level Gaussian distributions over the parameters with a mean vector $\boldsymbol{\mu}$ and standard deviations $\boldsymbol{\sigma}$. We initialised the group-level Gaussians as uninformative priors with means of 0.1 (plus some added noise) and variance of 100. During the maximisation step, we recomputed $\boldsymbol{\mu}$ and $\boldsymbol{\sigma}$ based on the estimated set of \mathbf{h}_i and their Hessian matrix \mathbf{H}_i (as calculated with Matlab's *fminunc*) over all N sessions. We repeated expectation and maximisation steps iteratively until convergence of the posterior likelihood NPL_i summed over the group or a maximum of 800 steps. Convergence was defined as a change in $NPL_i < 0.001$ from one iteration to the next.

4) The authors present results on the EEG response (heartbeat-evoked potentials) to signed (positive and negative) prediction errors. I would assume that positive and negative PE are highly correlated with outcome valence (reward/non-reward), so likely this presents a main effect of outcome? I would assume that this issue only pertains to signed, not to unsigned PE, but please report the correlations for both. If my assumption about correlation with signed PE is true, then these effects (figure 2B) should be interpreted as (most likely) reflecting outcome valence.

This is a very good point, and we understand the reviewer's reasoning. In classical learning tasks (Q-learning), outcome valence and signed PE are very highly correlated. However, in prediction tasks, like the one we employed, the stimuli's values need to be combined to predict an event (in our case a colour). In this case, the choice is made on the potential colour. In such tasks, PEs, in fact, only have limited covariation with the outcomes. This is illustrated below for an example participant across the four blocks. In the revised manuscript we have clarified and quantified the strength of the relation as shown below.

Caption. Illustration of relationship between outcome valence and signed PE for the four types of blocks used in the task.

5) In figure 3, the authors use regularized Fisher discriminant analysis ("machine learning") on the HEP across all electrodes to distinguish high vs low surprise trials (highest vs lowest quantile of absolute PE). To me, this recapitulates the mass-univariate results from figure 2 with a multivariate approach - which likely confers higher sensitivity. However, the difference between positive and negative PE which was found in figure 2B is not observed here. Do you have an explanation for that?

The reviewer raises an important point here. It is important to note that the regularised Fisher discriminant analysis and the mass-univariate analysis rely on fundamentally distinctive features of EEG data. The mass-univariate analysis focuses on discriminating amplitude changes of event-related potentials resulting from averaging all trials with positive vs. negative / high vs. low PE and computed on a relevant set of electrode sites - as informed by the topography analysis and coinciding with previous literature. By contrast, the machine learning approach capitalises on the trial-to-trial variability of the EEG data computed across all of the recording sites which allows us to accurately measure the changes in the HEP signal that fluctuate with time on a trial-by-trial basis whilst learning is taking place. Therefore, due to the distinctive fundamental properties of these two analysis approaches, it is possible that the significant differences between high and low absolute PE found when assessing trial-to-trial HEP changes reflect the contribution of interoceptive brain responses as learning unfolds. By contrast, while there may be differences in HEP amplitude once trials are categorised in a binary way as being high or low PE trials in the mass-univariate analysis, these differences may arise in a trial-by-trial and graded fashion in a manner that would be identified by the multivariate analysis approach. We have tried to address this in the manuscript.

Rather than inspecting specific electrode averages, we next wanted to search for EEG heart-beat features that predict the learning axes. We thus moved on to identify the whole-brain heartbeat-evoked neural components of learning by using a multivariate single-trial discriminant analysis of the EEG (regularised Fisher Discriminant analysis, see Methods) of the *HEP-locked signals*. More specifically, for each participant, we used the *average of the HEP* for each outcome (see Fig.3A) and calculated the linear weights associated with each electrode that maximally separated (1) positive and negative signed PEs (Fig.3E) and (2) high versus low magnitude of absolute PE (i.e. the size of the unsigned PE which describes how surprising the outcome is) (Fig.3A). We did this over multiple temporal windows and quantified the classification performance by using the area under the curve (Az) using a leave-one out approach. This method has been well established in EEG data analysis (Fouragnan et al., 2015, 2017; Komarnyckyj et al., 2022). Using this machine learning approach, we showed the presence of a large *heart-related* component reliably discriminating – even in individual participants (see Supplementary fig. 2) – between very high versus very low absolute PE outcomes. This component peaked in the time range 100–300 milliseconds after the R-wave (see EEG analysis for R-wave definition; Fig.3A). On the other hand, we did not observe any *heart-related* component discriminating between positive and negative PEs (Fig.F). By contrast, as noted, a grand average response difference *between positive and negative signed PEs* was identified in the ERP in fronto-central electrodes (Fig.2B). In conjunction, the two results suggest that there is, on average, a difference in cardiac related signals when the valence of the signed PE is positive or negative but that trial-by-trial variability in the *HEP does not reliably covary on a trial-by-trial basis with the trial-by-trial change in signed PE* (Fig.3F).

In summary, the different analysis approaches (ERP and machine learning) suggest the possibility of a number of relationships between *HEP* and learning signals but converge in suggesting an especially clear link, even at the trial-by-trial level, between the *HEP* and absolute PEs connoting how surprising or salient an outcome is. *We therefore focussed our analysis on the HEP component carrying absolute-PE information that we refer to as absPE-HEP. It is important to note that the regularised Fisher discriminant analysis and the mass-univariate analysis rely on fundamentally distinctive features of EEG data. The mass-univariate analysis focuses on discriminating amplitude changes of event-related potentials resulting from averaging all trials in a few electrodes. By contrast, the machine learning approach capitalises on the trial-to-trial variability of the EEG data computed across all the recording sites which allows us to accurately measure the changes in the HEP signal that fluctuate with time on a trial-by-trial basis whilst learning is taking place.*

An interesting finding in figure 3 is that high vs low surprise trials only differed locked to the 1st, but not 2nd or 3rd heartbeat during the outcome period. My understanding is that for the heartbeat-evoked potentials presented in figure 2, all heartbeats during the outcome period were used. If so, then the findings from figure 3D suggest to re-run this analysis including only responses locked to the 1st heartbeat. While post hoc, this may provide useful additional information.

This is a very good point. We have done this analysis and reported it in a new supplementary figure 5. When the analysis is performed in this manner, it continues to support the arguments made elsewhere.

Supplementary Figure 5. (a-b) stHEP waveforms across all trials following the onset of the R-wave (at time 0 ms) are shown separately for positive and negative signed PEs for the frontal cluster in (b) all HEP after feedback and (c) the first HEP after feedback and (d-e) high and low surprising outcomes (absolute PEs) for the central cluster in (d) all HEP after feedback and (e) the first HEP after feedback. The dotted line represents the difference between the conditions (represented in red and blue).

6) Figure 4: I have difficulty understanding this figure, it seems that what is presented does not fully match the description in the text or legend. My understanding is that, here, responses locked to the first heartbeat after outcome onset are split depending on whether the outcome occurred at systole vs diastole. Then it says (lines 303-306): "Having split the outcomes [...], we found that the associated absolute PE-related CRS depended on whether the outcome was presented at systole or diastole, with the mean CRS magnitude related to absolute PE being more negative when outcomes were presented at the diastole compared to the systole phase ($t_{31}=2.8460$, $p=0.0078$, fig4b)." I do not see where the absolute PE plays in here? To me, this indicates that heartbeat-locked responses during the outcome period are more negative during diastole vs systole, independent of outcome valence or surprise or anything else?

Thank you for the comment. Here we aimed to investigate the hypothesis that the relationship between the outcome onset and the cardiac cycle - i.e. outcomes naturally

appearing at either the systolic or the diastolic phase of the cardiac cycle, affects the brain interoceptive response (single-trial HEP) associated with high vs low absolute PE that we refer to as *absPE-HEP*. To this end we focused on the first HEP after outcome onset. In particular, we investigated if *absPE-HEP* was of high vs low value depending on whether the outcome was presented at systole vs diastole. The results showed that in trials where the outcome naturally appeared in diastole, the associated *absPE-HEP* was of high value, whereas the opposite was true when the outcome appeared during the systolic phase of the cardiac cycle. We realise that perhaps this was not clear in the previous version of our manuscript, so we tried to be careful and more precise when referring to the Machine Learning results.

We therefore focussed our analysis on the HEP component carrying absolute-PE information that we refer to as absPE-HEP. [...] To test whether the heart-related absolute PE (absPE-HEP) was parametrically modulated by the model absolute PE estimated in our model (rather than responding categorically to very high vs. very low absolute PE), we then calculated the discriminator amplitudes for trials with intermediate absolute PE levels (i.e. low absolute PE [0.25 – 0.50]; and high absolute PE [0.5 – 0.75]) which were not originally used to train the classifier (also called the “unseen” data). To do so, we applied the spatial weights of the peak discrimination performance for the extreme outcome absolute PE levels to the EEG data with intermediate values. We expected that the discriminator amplitudes for these previously “unseen” trials would increase linearly as a function of absolute PE. Thus, the resulting mean amplitude at the time of peak discrimination would proceed from very low < low < medium < very high absolute PE. This is indeed what we found (Fig. 3c, blue: intermediate categories, grey: categories used for discrimination) confirming the linear relationship between the absPE-HEP component and its model-based counterpart (test on the left-out data: $t_{31} = -7.3027$; $p = 3.22e-8$; $CI = [-0.1187 -0.0669]$, Cohen $d = -1.129$) and also the generalisability and robustness of our machine learning approach. Having applied the estimated electrode weights to single-trial data to produce a measurement of the discriminating component amplitudes (representing the distance of individual trials from the discriminating hyperplane), we thereafter used these amplitudes for all subsequent analyses involving absPE-HEP.

Jumping to figure 4E, this seems to speak to the above conclusion, however, I am again not sure whether this indeed provides evidence for a modulation of PE-responses by cardiac cycle: the authors describe a main effect of cardiac cycle and a main effect of signed PE - but no interaction. Thus, to me this describes that heartbeat-locked EEG outcome effects are (i) sensitive to high vs low surprise (as shown in figures 2/3) and (ii) are more pronounced during systole vs diastole (as shown in figure 4B), but not that PE-related responses are modulated?

We hope the answer to the previous question answers this question as well. In addition, as the figure has changed, this panel has now been removed to present instead the analysis suggested by the reviewer. Note that the full statistical result for this analysis is reported in the text.

Please also add error bars to 4E.

As the figure has changed, this has now been removed.

What does figure 4C show, is it the same as 4E, but only for signed PE near zero?

Apologies if this was not clear. Fig.4C represents the near threshold *absPE-HEP* (*absPE-HEP* < 50% percentile). We hope we have been able to clarify the reading and the figure.

Figure 4D does not seem to be referred to in the text.

We have now added this in the text, apologies for the original missing point:

The two phases were also associated with similar levels of overall reward received in the task ($t_{31}=0.8046$, $p=0.4272$, $CI=[-0.008\ 0.02]$, $Cohen\ D=0.14$, supplementary Fig.3b), unsigned PEs from the RL model ($t_{31}=-0.0581$, $p=0.9541$, $CI=[-0.026\ 0.025]$, $Cohen\ D=-0.01$, Fig.4d) or signed PE ($t_{31}= -0.72$, $p= 0.4769$, $CI=[-0.042\ 0.02]$, $Cohen\ D=-0.12$). This is important as it shows that the cardiac cycle does not modulate model estimates in a direct way.

Figure 4F-K, the legend does not describe very well what is shown in the plots.

We have now changed this.

Figure 4. Influence of the cardiac cycle on the *absPE-HEP* and learning. **(a)** Schematic description of the systole and diastole phases. In red and blue are the systole and diastole periods respectively. Below is a representation of two example trials on which outcome onsets happened at systole or diastole. *We then looked at the EEG response locked to the next heart-beat.* **(b)** Mean amplitude difference between the *absPE-HEP* of the first heartbeat after all outcomes presented at systole versus diastole ($N=32$). A violin plot is used to present all individual participants' averages. **(c)** Mean amplitude difference in the *absPE-HEP* for the low salient outcomes presented at systole versus diastole across subjects ($N=32$). **(d)** Mean amplitude difference in *absPE-HEP* for the high salient outcomes presented at systole versus diastole ($N=32$). **(e)** A logistic regression analysis showed that switch / stay (1 / 0) could be predicted by several predictors, including the Absolute PEs

(participants were more likely to switch after a highly surprising outcome) but also residual absPE-HEP. (f-k) Results of the correlation between the regression coefficient for each participant between absPE-HEP and systole/distole and the mean reward and learning rates in the task. In red, the fit of the robust regression. Any of these results remain true even when including a covariate indexing features of the external outcome type – reward and absolute PE from the model. Particularly, in (f) learning rates – all task blocks (g) learning rates – predictive blocks (h) learning rates – non predictive blocks (i) reward – all task blocks (j) reward - predictive blocks (k) reward – non predictive blocks.

Minor:

1) The literature is not cited carefully. I find it alarming that of the six references I checked, none was actually a study that related to the findings cited in the text: lines 430-432: "Neuronal excitability is influenced by the cardiac cycle; whilst neural signals from the baroreceptors occurring at systole attenuate concurrent brain activity (12, 13), and impair information processing, enhanced excitability and perceptual processing is observed during diastole (14, 15)" <-- Ref 12 is the first authors own work, 13 Leong et al., none of the two studies have reported effects related to the cardiac cycle. 14 is Hayden's ACC foraging study, 15 one of Schultz' many dopamine reviews, again (to my knowledge) nothing on cardiac cycle and neural excitability. Likewise, (line 447) ref 19 (Kahnt & Tobler) does not measure "cardiac interoceptive sensitivity" and (lines 450-453) ref 20 (Behrens et al. 2007) is not at all about "Increased and decreased cardiac sensitivity has also been shown to help or hinder adaptive intuitive decision making..." Please carefully check all references.

Thank you for your thorough reading through the manuscript. We are sorry there were issues with the referencing. We have gone through the whole manuscript and believe these are all fixed. The discussion now reads:

Importantly, we also observed that the influence of the cardiac cycle on the absPE-HEP magnitude progressively increased as the outcome absolute PE became smaller. In outcomes with near-threshold absolute PEs, the absPE-HEP magnitude increase was predominantly observed when the outcome was presented at diastole (Fig.4). This means that when the decision maker's prior expectations are close to the outcome (i.e. small adjustments between expectations and outcomes) learning is more likely to occur during the quiescent phase of the cardiac cycle than during the active, systolic phase. Neuronal excitability is influenced by the cardiac cycle; whilst neural signals from the baroreceptors occurring at systole attenuate concurrent brain activity (Duschek et al., 2013; Lacey & Lacey, 1970) and impair information processing, enhanced excitability and perceptual processing is observed during diastole (Al et al., 2020; Walker & Sandman, 1982). Formally, enhanced neuronal excitability may increase neural gain, which directly translates into an increase of the breadth of attention towards the aspects of the environment to which one is predisposed to attend (Eldar et al., 2013). Here we show that in instances where learning happens in small increments because the PE-related surprise

is not very salient, learning is enhanced during diastole compared to systole, helping to update prior expectations even when there is little new information available.

2) I would recommend not to refer to near-zero (absolute) PE as "near-threshold" - it may be misleading, and for some readers it might imply something about the stimulus being near perceptual threshold.

We take note of the reviewer's point but think that this is perhaps best addressed by very carefully explaining what we mean by near threshold in the current context; in the revised manuscript we add an additional clarification to remind the reader that "near threshold" in the context of the revised manuscript refers to the PE being near the threshold for learning rather than that feedback is near the threshold for visual detection.

3) There is little behaviour in the results, primarily modelling. I think supplementary figure 1C shows learning curves, which I'd recommend to include in the main results. Looking at this curve, there seems to be surprisingly variable and near-chance performance in many blocks, barring a few exceptions.

Apologies if this was not clear. Supplementary figure 1 shows choices and not learning curves. We agree with the reviewer that the choices on the different blocks can look random even if the learning has happened. For example, in the anticorrelated blocks, the two cues are strictly predictive of two different colours. So, if a face and house are both presented in a single trial, there is a 50% chance of getting one of the colours or the other. In this case the correct answer is to flip a coin between the two colours so the choices will look random.

4) Fig. 4E, y-axis label, what is STV?

Single-trial variability. This panel in this figure has been removed though.

Reviewer #3:

I read this report closely, and I have a mixed opinion of the conclusions. The authors demonstrate that prediction error learning precision is enhanced (i.e. better learning to low-information feedback) when feedback is timed to diastole in the cardiac cycle. Much of my hesitation is due to one major concern that needs to be addressed: there are a lot of reasons to be skeptical of heartbeat-evoked EEG activities. There were some other rather minor issues that could be addressed to help diminish the complexity of this report.

Major Concerns

1) The cardiac-related neural signal (CRS) or heartbeat evoked potential (HEP) is a controversial method. The authors note the methodological difficulty of linking the CRS since the cardiac field artifact (CFA) can be averaged into this potential. My concern is not with this electrical contaminant and whether or not it was removed via ICA. My concern is with the bolus-related movement of tissue or sensor, which is a possible separate artifact that is often overlooked in this field. I have followed a lot of work in this field, and I have never found this issue adequately addressed. I think there is a high probability that the slow peak and decline $\sim 100\text{ms} - 600\text{ms}$ following the R-spike is due to stereotyped movement of tissue or sensor with the pulsed blood flow, which becomes averaged together into a voltage change. For analogy, you could create a movement-related potential if you had a sensor on a grapefruit and you reliably tapped it in sync with a timing trigger. I think that addressing this potential confound is a prerequisite to interpreting a CRS.

The reviewer raises an important point here, one that we have also carefully thought of but perhaps failed to report carefully. We have now revised the reporting of our methods accordingly and added a new section on this potential confound.

We believe that if the data were contaminated by such artefact, it would very likely arise due to the motion of EEG electrodes because of local pulsatile movement of the scalp during the cardiac cycle, due to varying blood flow through scalp vessels during cardiac rhythms. Such a mechanism is, indeed, not often discussed but it is a possibility. We are aware of its possibility because it is actually very similar to a better studied phenomenon, known as the **ballistocardiogram (BCG) artefact**, which we have previously investigated in our simultaneous EEG-fMRI work (Fouragnan et al., 2015 Nature Communications; Pisauro et al., Nature Communications 2017; Fouragnan et al., 2017 Scientific Reports; Arabadzhyska et al., 2022 Journal of Neuroscience) which arises due to the motion of EEG electrodes in the static magnetic field of the fMRI scanner. Obviously, the BCG is an extreme version of such an artefact, but we believe that the possible artefact that the reviewer notes and the BCG share common features. Stereotyped movement of tissue or the sensor with the pulsed blood flow of the BCG artefact are characterised by their **temporal relation** with the cardiac rhythms captured by electrocardiogram. Several methods have been used to deal with these artefacts but one of the most successful methods is, in fact, ICA removal. ICA would not only detect heart-related electrical contaminant but any other contaminant with a frequency identical to

the heart pulse rate, for example an artifact related to pulsatility. We appreciate that this might not be obvious and welcome the opportunity to make this point clearer.

We try to explain this argument as follows. After low-pass filter at 4 Hz to extract the signal within the frequency range where usually the pulse related artifact artefact is observed, and where any artefact due to bolus-related movement of tissue is expected to occur, we typically identify several independent components representing the artefact. These explain most of the variance in the evoked BCG signals in a specific time range. The method that we implemented in the manuscript, that we referred to as ECG correction, therefore, automatically picks out any feature that might be caused by a pulse affecting the electrodes. Although this may not have been obvious in the first version of the manuscript, we have tried to explain in the method.

Automatic artefact rejection was performed excluding trials and channels whose variance (z scores) across the experimental session exceeded a threshold of 20 μ V. This was combined with visual inspection for all participants eliminating large technical and movement related artefacts. Physiological artefacts such as eye blinks, saccades and the volume-conducted cardiac-field artifact (CFA) were corrected, in all participants, by means of independent component analysis (RUNICA, logistic Infomax algorithm) as implemented in the FieldTrip toolbox. Importantly, the data could also be contaminated by stereotyped movement of tissue or sensor with the pulsed blood flow, which becomes averaged together into a voltage change. Stereotyped movements of tissue or sensors with the pulsed blood flow are characterised by their temporal relation with the cardiac rhythms captured by electrocardiogram. Several methods have been used to deal with these artefacts but one of the most successful methods is, in fact, ICA removal (Fouragnan et al., 2015, 2017; Srivastava et al., 2005). Those independent components (4.78 on average across participants; 1.13 SD) whose timing and topography resembled the characteristics of the physiological artefacts were removed. The CFA represents a challenge to the analysis of the HEP because the averaging of the data around the R-peak amplifies the CFA that are time-locked to the heartbeat (Luft & Bhattacharya, 2015). Nonetheless, ICA has been shown to be of high efficiency in the removal of the independent components representing CFAs from the EEG signal (Park et al., 2014, 2017; Terhaar et al., 2012). The IC identification and selection process were guided by visual inspection of their properties, based on time course and scalp topography. ECG channels were excluded from the analysis and the signal was then re-referenced to the arithmetic average of all electrodes.

Importantly, we would like to highlight that our multivariate analysis detects task-related discriminating components which are likely to be orthogonal to these artifacts and particularly pulsing artifacts (Fouragnan et al., 2015; Gherman and Philiastides, 2018). In fact, in previous EEG-fMRI studies, we have found that our approach is robust to the presence of pulsing artifact residuals, specifically, because of the (spatial) multivariate nature of our classification techniques.

In addition, it may reassure the reviewer to note that we also considered the possibility of other forms of spurious, movement artefacts including jaw clenching, eye blinking, BCG, and EMG from finger and hand muscles at the time of responding. We removed these artefacts from the data whenever they were found. Jaw clenching and eye blinking were the most frequent causes of artefacts.

Finally, as a last precaution, we tested for any relationship between heartbeat rates as indexed by the interbeat interval (IBI) on each trial, and behavioural measures. We found no relationship between IBI and PE or $|PE|$ and neither for the interindividual variation in IBI and the results presented in Figure F-K.

2) The authors do show in Figure 4f-h that learning depends on CRS features, and only in learnable blocks. Yet this still could be an artefact of known effects of cardiac period on information intake (see minor point #2): the cardiac phase could influence learning and it could influence EEG, but EEG findings could be mediating or they could be spurious (i.e. major point #1 above). If the authors could demonstrate that this behavioural effect is intimately related to the findings shown from the CRS (e.g. mediating not spurious), then that would be very helpful for addressing this concern.

In the revised manuscript we present a new analysis that speaks to this issue. Previously, in the initial report, we showed that the HEP reflected the size of the unsigned prediction error (unsigned PE). We also know that the unsigned PE is one determinant of the degree to which participants update their decision (and we showed that this is the case in the current study too). However, one way to demonstrate that HEP is “intimately related” or “mediating” behavioural change is to look at the residual variance in the HEP that is unexplained by unsigned PE and test whether this also explains the changes in behaviour that are seen in the next trial.

To do this we tested whether the single-trial HEP reflected the size of the unsigned PE (obtained from the machine learning-based analysis) at outcome (t) could predict change in choices in the next trial at t+1. To do so, we ran an additional regression to predict choice switching behaviours at t+1 with the residual single trial HEP variance at t (after controlling for the part of the HEP that was collinear with the absolute PE) and added the systole and diastole as a separate regressor. Our results showed that participants were more likely to switch after a highly surprising outcome (absPE: $p = 1.4791e-13$, tstat: 12.4041, df: 31) aligning with previous reports (Fouragnan et al. 2017 Scientific Reports) but also that higher residual variance in HEP, even after controlling for that part of the HEP that was collinear with unsigned PE, also predicted switch behaviours on the next trials (absPE_HEP: $p = 0.0162$, tstat: -2.5440, df: 31). In addition to this key result, the analysis, also revealed that there was a trend towards significance suggesting that the cardiac cycle on the previous trials (whether the outcome’s onset happened at systole or diastole) also predicted a switch (i.e. whether the outcome onset occurred at diastole or systole on trials prior to t tended to predict choice

switching on trial t+1: SysDias: $p = 0.0591$, $t_{stat} = -1.9592$, $df = 31$). Interestingly, this seems to align with our previous results showing that salience-related HEP at outcome are stronger at diastole than systole. In a similar way, the outcomes falling at diastole tend to predict switch behaviours more strongly than outcomes falling at systole. This analysis does not show any interaction effects. By contrast other aspects of behaviour unrelated to learning such as reaction times RTs) could not be predicted except by absPE: $p = 0.0120$, $t_{stat} = 2.6684$, $df = 31$. Thus, to summarise, this approach to analysing the data reveals a number of interesting results. Perhaps the key one, that most directly addresses R3's request, is that residual variance in HEP, even after controlling for that part of the HEP that was collinear with absolute prediction error, also predicted switch behaviours on the next trials.

Figure 3. Left: A logistic regression analysis showed that switch / stay (1 / 0) could be predicted by several predictors, including the Absolute PEs (participants were more likely to switch after a highly surprising outcome) but also residual cardiac related signals. This figure is now presented on Figure 4 of the main manuscript. **Right:** This was not the case for a linear regression model predicting Response Times.

We are now presenting this in the new Figure 4, panel E:

Figure 4. Influence of the cardiac cycle on the *absPE-HEP* and learning. **(a)** Schematic description of the systole and diastole phases. In red and blue are the systole and diastole periods respectively. Below is a representation of two example trials on which outcome onsets happened at systole or diastole. We then looked at the EEG response locked to the next heart-beat. **(b)** Mean amplitude difference between the *absPE-HEP* of the first heartbeat after all outcomes presented at systole versus diastole ($N=32$). A violin plot is used to present all individual participants' averages. **(c)** Mean amplitude difference in the *absPE-HEP* for the low salient outcomes presented at systole versus diastole across subjects ($N=32$). **(d)** Mean amplitude difference in *absPE-HEP* for the high salient outcomes presented at systole versus diastole ($N=32$). **(e)** A logistic regression analysis showed that switch / stay (1 / 0) could be predicted by several predictors, including the Absolute PEs (participants were more likely to switch after a highly surprising outcome) but also residual *absPE-HEP*. **(f-k)** Results of the correlation between the regression coefficient for each participant between *absPE-HEP* and systole/distole and the mean reward and learning rates in the task. In red, the fit of the robust regression. Any of these results remain true even when including a covariate indexing features of the external outcome type – reward and absolute PE from the model. Particularly, in **(f)** learning rates – all task blocks **(g)** learning rates - predictive blocks **(h)** learning rates - non predictive blocks **(i)** reward – all task blocks **(j)** reward - predictive blocks **(k)** reward - non predictive blocks.

Minor Concerns

1) I spent a lot of time double checking what condition the absolute PE was derived from. I initially (correctly) assumed it was the absolute value of the feedback (+ or -) PEs. Yet Fig 1h led me to consider that the second stimulus presentation could be framed as an absolute PE since after the first stimulus it is unknown what the outcome will be, but the second stimulus will potentially resolve this uncertainty (with a salience PE). Thus I spent time understanding

if the data were locked to that event, not the feedback. Figure 3a and 3e were helpful to resolve this confusion, but I think an alteration of 1h would be helpful. It would be helpful to have a very clear sentence in the text that formally defines absolute PE (line 240 helps, but it's a little bit of info, a little late).

This is a very good point; we have revised figure 1 to make clear that the activity that we examine is time-locked to the outcome event. We now emphasise that the two quantities, PE and absolute PE, are examined at time of feedback and associated with variation in largely distinct distributed neural representations which carry separate roles (Fouragnan et al. 2017, Fouragnan et al. 2018, Queirazza et al. 2019). One of these roles is to drive the organism towards allocating more attention and, as a further consequence, this may lead to greater future adjustments in the organism's estimates of choice values. We amended the introduction [pg. 3] and we modified the references to correctly reflect this point. We have now changed the legend in our first figure to reflect this point.

Figure 1. Schematic representation of the task and RL results for all four association schemes, highly predictive anticorrelated (HA), highly predictive correlated (HC), variable predictive (VP) and non-predictive (NP). **(a)** task + cardiac-related neural signals (stHEP) recorded at outcome for 4 sec - enough to get on average 3.5 HEPs **(b)** example of association between cues and predicted colour for one version of the task (anti-correlated blocks – see Methods) **(c)** prediction strengths for each block **(d)** model prediction of choices **(e)** Learning rates from the best fitting model **(f-g)** Reaction times and performance (mean reward across the four blocks **(h** Left panel) Definition of absolute and signed PE (right panel) Each coloured pattern illustrates a different way in which activity in a neural structure may manifest if it is sensitive to different aspects of outcomes and their associated PE (locked at time of feedback). The blue pattern illustrates activity as a function of outcome valence – in a categorical manner as either positive or negative. The red pattern shows a monotonically increasing response profile

consistent with a continually varying PE representation. The black pattern is continually varying, as a function of the difference between the outcome and the expectation in an unsigned fashion (i) Signed PEs are presented across participants for all blocks, following the example of a task like in Fig1C (j) variation of PEs across blocks (k-l) similar to (i-j) for absolute PEs.

Note that we are not looking at the absolute value of the feedback but the absolute value of the PE. In classical learning tasks (Q-learning), outcome valence and signed PE are very highly correlated. However, in prediction tasks, like the one we employed, the stimuli's values need to be combined to predict an event (in our case a colour). In this case, the choice is made on the potential colour. In such tasks, PEs, in fact, only have limited covariation with the outcomes.

2) There is old literature in psychophysiology on cardiac slowing during “information intake”. The work from the Laceys and Obrist comes to mind, as well as some critical re-appraisals (see below). If the authors are interested, this would be a very interesting old corpus to integrate with this new approach.

(<https://www.sciencedirect.com/science/article/abs/pii/0301051178900595>,
<https://www.sciencedirect.com/science/article/abs/pii/0301051180900277>,
<https://psycnet.apa.org/record/1973-10545-001?doi=1>,
<https://onlinelibrary.wiley.com/doi/abs/10.1111/j.1469-8986.1970.tb02257.x>)

We thank the reviewer for the insightful suggestions. We have incorporated these ideas in our manuscript in the Introduction and discussion sections.

In the introduction it reads as follows:

*Learning is affected by states of cognitive and physiological arousal that can fluctuate over time (Liu et al., 2015). For example, it is long established that heart rate slows down in situations such as learning that require attention to the environment (Lacey & Lacey, 1970). Although the exact functional role of cardiac deceleration on cognition is still under debate (Green, 1980; Hahn, 1973), heart rate deceleration - which involves longer diastolic phase, is appreciated as a physiological mechanism that better prepares the organism to take in sensory stimuli and respond to them (Obrist et al., 1970). Our aim in the current study is to examine the relationship between the cardiac cycle and quantitative indices of the learning process such as signed and absolute PEs. Importantly, model estimates of signed and absolute PE can capture **cognitive and physiological** fluctuations as learning progresses.*

In the discussion, it reads as follows:

*Neuronal models of interoception conceptualise cardiac predictions as afferent signals projecting to agranular visceromotor areas in frontal **cortex** and anterior insula **cortex**, which serves as the primary interoceptive cortex (Evrard et al., 2014; Saleem et al., 2008). The anterior insula is argued to be a main neural source for the **HEP** along with other interconnected areas such as the cingulate and the somatosensory cortices (Babo-Rebelo et al., 2016; Kern et al., 2013; Park et al., 2017). These brain regions belong to a wider network, often referred to as the salience network, which is sensitive to homeostatically relevant*

stimuli independent of whether their valence is negative (penalising) or positive (reinforcing) (Bartra et al., 2013). It is becoming increasingly clear that neural responses in the absolute PE network rise quickly after an outcome is revealed (Algermissen & Ouden, 2022; Fouragnan et al., 2015). Here we observe that the HEP is parametrically modulated by the outcome's absolute PE and that this is mainly due to the first heartbeat recorded immediately after the outcome onset. This means that HEP magnitude changes recorded immediately after outcome can be used as a proxy for attentional allocation to the internal representation of absolute PE. Our results are not inconsistent with a long-established notion that cardiac deceleration – i.e. longer diastolic phases, serves to make the organism better equipped to intake sensory stimuli and respond to them (Obrist et al., 1970).

Reviewer #4:

I would like to thank the editor for inviting me to review this interesting work. I congratulate the authors for the originality and methodological rigour. Still, it is not straightforward how to place these new findings among the existing literature. I recommend a revision on the results interpretation and a contextualization with respect to recent literature.

We thank the reviewer for highlighting that our study is interesting, original and methodologically rigorous. Based on the reviewer's comments (see below for details) we now amended the manuscript to better highlight the novelty of this work. As a result, we hope that the manuscript is clearer.

My specific points below:

1. In the introduction's 1st and 3rd paragraphs it seems that the authors agree on a binary view of perception, depending on the cardiac cycle phases. However, abundant evidence exists to the date showing that these heart-brain interactions are complex and non-linear phenomena. As described for instance in Silvani et al (10.1098/rsta.2015.0181) and Candia-Rivera et al (10.1016/j.crneur.2022.100050). Furthermore, in Skora et al (10.1016/j.neubiorev.2022.104655) it is reviewed that this cardiac phase hypothesis does not go in the same direction in all reported studies. Indeed, in the results (Figure 4) what is shown is a trend.

We thank the reviewer for the helpful comment. It was not our intention to suggest that large changes in binary classification of trials, as ones on which stimuli are perceived with certainty or not perceived with certainty, occur as a function of cardiac cycle. We acknowledge that the effects are small and non-linear. In brief, as suggested we now refer to, and comment on, the complexity of the heart-brain interactions incorporating some of the relevant literature. The changes in the manuscript to incorporate the reviewer's suggestions read as follows:

In situations where we must make decisions based on noisy or incomplete information - for example deciding whether to cross the street on a foggy morning with poor visibility - our choices can be modulated, albeit to a small degree, by the timing of the cardiac cycle. Studies investigating near-threshold sensory events, like visual, auditory or somatosensory events, have shown that timing in the cardiac cycle (e.g., whether events happen during the systolic or diastolic phases of the cardiac cycle) impacts the perception and reaction to

sensory cues through changes in associated neural signals (Al et al., 2020; Candia-Rivera, 2022; Motyka et al., 2019).

... Behavioural and neuroimaging research suggests that sensory perception and executive control are affected *differently* by the heart phase (Al et al., 2020; Sandman, 1984; Walker & Sandman, 1982). Although such a distinction remains debated (Skora et al., 2022), the studies show that participants are more sensitive to *perceiving visual, auditory and somatosensory* signals during diastole and less sensitive during systole when key sensory brain regions receive cardiac related afferent signals increasing the excitability levels in these regions. *By contrast, executive processes such as attention switching, active sampling and motor control may be enhanced during systole as opposed to diastole. This suggests that different cognitive processes may be prioritized at different points in the cardiac cycle* (Candia-Rivera, 2022; Silvani et al., 2016).

2. How these results stand with the literature on decision and heart-brain interactions? The authors limit their comments in the discussion to general literature on interoception, and with respect to Galvez-Pol 2020, 2022 studies. I believe the discussion could be enriched in this regard. Here are some studies showing correlates between heart-brain interactions and different dimensions of decision-making:

- preference consistency (Azzalini et al, 10.1523/JNEUROSCI.1932-20.2021),
- response to stimuli (Larra et al, 10.1038/s41598-020-61068-1),
- action execution (Palser et al, 10.1016/j.cognition.2021.104907),
- response inhibition (Rae et al, 10.1038/s41598-018-27513-y; Ren et al, 10.1016/j.biopsycho.2022.108323),
- choice-reward (Fujimoto et al, 10.1073/pnas.2014781118)

Thank you for the insightful comment. We have added the suggested references in the Introduction:

Although heart-brain interactions are starting to be understood in relation to sensory-driven processes, it is unclear whether the cardiac cycle has a similar impact on other internal representations which are non-sensory but which, like sensory stimuli, shape decision making (Azzalini et al., 2021; Fujimoto et al., 2021; Larra et al., 2020; Palser et al., 2021; Rae et al., 2018; Ren et al., 2022). Here we focus on a much-studied internal representation – the reward prediction error [PE] (Schultz, 2016) – and investigate whether the cardiac cycle also determines the impact that each PE will have on learning. Importantly, the magnitude of the PE can be dissociated from the magnitude of the accompanying sensory stimulus. This makes it possible to determine whether the cardiac cycle has an impact on near-threshold PEs even if the PEs are associated with clearly suprathreshold sensory stimuli.

And in addition we have made the following changes to the manuscript in the Discussion:

Our results demonstrate that the **HEP** recorded during the presentation of reward-related outcomes discriminates between different levels of absolute PE outcome. By contrast, the magnitude of the **HEP** did not differ when contrasting positive versus negative signed PEs. The absolute PE and signed PE components of reward learning subserve different functional roles in learning (Fouragnan et al., 2017, 2018; Rouhani & Niv, 2021); whilst signed PE is associated with approach-avoidance behaviour, absolute PE, also called salience impacts on future attentional engagement; an effect that is determined by the magnitude of the discrepancy between prior expectations and outcome. **During learning, we found that single-trial HEP sizes were also related to absolute PE sizes when decision outcomes occurred. Moreover, some of the variation in the absPE-HEP also predicted whether participants were more likely to shift to a different decision when the decision they had just made had led to a surprising outcome. In other words, the absPE-HEP did not predispose participants to make one choice or another but larger absPE-HEP were associated with surprising feedback information having a bigger impact on changing the way that subsequent decisions were made.**

This effect of cardiac cycle on learning is analogous to some cardiac-related effects that have been reported in the context of decision making. For example, cardiac responses in decision-related brain areas such as ventromedial prefrontal cortex are larger when the decision-related information will have a bigger impact on the decisions made (Azzalini et al., 2021). Both in the decision-making result previously reported and in the current study, neural responses to the cardiac cycle are related to how impactful concurrent information will be on behaviour rather than with a particular type of behaviour. Similarly, single neuron responses recorded in adjacent orbitofrontal and anterior cingulate brain areas in macaques also vary with heart rate and heart rate is associated with a general increase in the speed of decision making (Fujimoto et al., 2021).

Cardiac neurophysiological responses often convey not only information about the current bodily state, but they also carry predictions of how the bodily system should organise internal resources to deal with expected future sensory information (Barrett & Simmons, 2015; Gentsch et al., 2019). These cardiac predictions are often accompanied by a modulation of attentional responses to upcoming stimuli that, ultimately, are homeostatically relevant. In this way, it has been suggested that the internal bodily state determines perceptual stimulus salience in relation to homeostatic levels (Barrett & Simmons, 2015; Paulus et al., 2009). For example, a stimulus occurring when resources are sparser may be perceived as more salient than a stimulus occurring when more resources are available. **The absPE-HEP might signal that more attention needs to be deployed to the current outcome given the current bodily state. In this way a bodily signal might modulate learning.**

[...]

Previous studies have carefully time-locked the presentation of stimuli to the cardiac phase to investigate differences in the way **that stimuli** are processed (Al et al., 2020; Larra et al., 2020; Skora et al., 2022). **For example, tactile stimuli presented at diastole are more frequently detected than those presented at systole (Al et al., 2020). Conversely, the ability**

to control movements is facilitated during cardiac systole (Rae et al., 2018) – albeit this tendency reverses when emotional cues are present (Ren et al., 2022). However, sensory or learning information is not presented in such a phase-locked manner in our everyday lives. By investigating how participants naturally receive information relevant for learning and assign credit for outcomes to objects maintained in memory, with respect to the natural timing of the cardiac cycle, we have adopted an ecological approach to studying brain-heart interactions in the context of learning and decision making. Previous studies, adopting a similar approach, have shown that people actively seek information in the world, or more precisely sample the world through active sensing, as a function of the cardiac cycle. For example, in an active sampling visual paradigm, saccades and visual fixations are more likely to occur in the quiescent phase of the cardiac cycle (e.g. diastole) (Galvez-Pol et al., 2020). Similar work suggests that people actively adjust sensory sampling so that more time is spent in the diastole period in which perceptual sensory sensitivity is enhanced (Galvez-Pol et al., 2022). Moreover, in dyadic interactions actions are more likely to take place during diastole, and also the observer is less likely to experience a heartbeat (systolic phase) when observing movement endpoints (Palser et al., 2021). In our study, we have shown that the magnitude of the single trial HEP is stronger when the outcome appeared during the diastole period in comparison to the systole period (Fig.4). This suggests that the phase of the cardiac cycle is an important modulator of internal representation and cognition and influences the way in which we naturally receive information.

3. Did the ICA computation include the ECG channel? If yes, how many leads? In the HEP literature exists a high heterogeneity on this preprocessing step, and therefore, it needs to be clarified. To exemplify, in some studies like in Couto et al (10.1016/j.autneu.2015.06.006) the CFA is not removed, in others like in Al et al (10.1073/pnas.1915629117) the ECG is not included in the ICA computation and CFA were removed only when found. In our recent guideline paper (10.1016/j.jneumeth.2021.109269) we recommended to include the ECG on the ICA computation, in addition to a final EEG average re-reference, as the authors apparently did.

Thank you for the relevant comment. The ICA computation indeed included the ECG channels. One channel was included because of our experimental set up. During the recording of the data, we used one external electrode to record heartbeats and this external electrode was embedded as part of the EEG recording system. We also conducted the re-reference to the average signal at the end of the preprocessing pipeline. In the revised manuscript we have pointed out that we have followed the procedure recommended in (10.1016/j.jneumeth.2021.109269).

4. The authors report on the methods, ICA part: “Those independent components (4.78 on average across participants; 1.13 SD)”. Is this the component number as sorted in Fieldtrip? Or is it the number of components?

The number reported is the average number of independent components removed, on average, across participants. We have revised the manuscript to make this clearer.

5. Did the authors remove the CFA in all participants? If not, please report the number of participants in which the CFA was not found.

Thank you for the comment. The method employed was successful at removing most of the noise coming from the CFA in all our participants. We have made this clearer in the revised manuscript.

6. Is there a reason to use the acronym CRS? Why not just HEP or HEP with a subscript?

We have changed the acronym CRS to HEP for simplicity.

7. Could you clarify on the methods how many epochs (or trials) are being averaged in each CRS? This could be relevant if only one heartbeat is considered after the outcomes. If it is only one epoch, it needs to be clearly justified and discussed whether it is a reliable measurement.

We have tried to examine this issue carefully. For all analyses done on average during the whole period after outcome, we have on average 3.4HEP after outcome. However, we have also realised that it is the cardiac-related neural activity that occurs with the first heartbeat after the outcome is presented that best carries absolute PE-related information. Including additional heartbeats in analyses does not lead to a change in results but it seems that the effect is mainly due to the first heartbeat-related activity.

Thus far, we have demonstrated that a large temporal component locked at R-wave onset, the biggest wave generated during normal heart conduction (see Methods for full definition) for all heart-evoked signals that were collected during the feedback period link with the absolute PE. However, the extent to which this temporal component was driven by the first, second, or third heartbeat that occurred in the outcome period remained unknown. Next, we therefore repeated our multivariate analysis independently for each of the three possible HEP times post-outcome, sequentially, across all trials to better understand the temporal dynamics of the absPE-HEP modulation. This approach allowed us to determine which heartbeat was most related to absolute PE. Applying this method, we showed that only the first HEP after feedback contained information about absolute PE that could be revealed with machine learning techniques, in the range 100–300 milliseconds after heart beat onset (Fig. 3D). This finding indicates that the first heartbeat after outcome is the one that relates most to the representation of the absolute PE of the outcome, suggesting that the timing of the outcome with respect to the cardiac cycle might be important in determining how participants update their internal representations of decision outcomes. Because our results highlight the importance of considering the first HEP after feedback rather than averaging all HEP after feedback, we also decided to redo our initial ERP analyses using only the first HEP after feedback albeit a lower statistical power. This is presented in supplementary figure 5.

REVIEWERS' COMMENTS

Reviewer #1 (Remarks to the Author):

The authors have worked hard to address my previous concerns and questions I had about this manuscript. I am happy with the paper and have no further comments

Reviewer #2 (Remarks to the Author):

The authors have performed a thorough revision. In particular they have added novel analyses that substantially strengthen their case, in addition to modifications to the manuscript that helped to clarify many points.

I was particularly convinced by the analyses displayed in new figure 4 E.

The only point I had not been entirely convinced by is the recovery of the computational model parameters, which are presented in supp. table 2 (supp. table 1 is somewhat redundant, no? table 2 shows the same as table 1 + recovered values). The recovered learning rates are (while still rather high) consistently lower than the ones fitted to behaviour. The differences in inverse temperature are even more striking, with recovered temperature being up to 10 x as high as the fitted values. I also think rather than presenting mean values, it would be more helpful to provide scatterplots and correlation coefficients in the supplements. I would like to emphasize that I think it is unlikely this would change the results a lot, as even somewhat deviant learning rates will likely lead to rather similar estimates for PE and $|PE|$, but it would still be helpful to have this information (scatter diagrams and correlation coeffs) presented.

One last suggestion, for the new computational models, the model parameters are only described, maybe you want to consider presenting the equations for ease of understanding?

Reviewer #3 (Remarks to the Author):

The authors have successfully addressed all of my concerns.

Reviewer #4 (Remarks to the Author):

The authors have successfully addressed all my comments and I recommend their article for publication

REVIEWERS'

COMMENTS

Reviewer #1 (Remarks to the Author):

The authors have worked hard to address my previous concerns and questions I had about this manuscript. I am happy with the paper and have no further comments.

Reviewer #3 (Remarks to the Author):

The authors have successfully addressed all of my concerns.

Reviewer #4 (Remarks to the Author):

The authors have successfully addressed all my comments and I recommend their article for publication.

We thank the three reviewers for their comments and recommendations.

Reviewer #2 (Remarks to the Author):

The authors have performed a thorough revision. In particular they have added novel analyses that substantially strengthen their case, in addition to modifications to the manuscript that helped to clarify many points.

I was particularly convinced by the analyses displayed in new figure 4 E.

The only point I had not been entirely convinced by is the recovery of the computational model parameters, which are presented in supp. table 2 (supp. table 1 is somewhat redundant, no? table 2 shows the same as table 1 + recovered values).

We have now implemented the changes suggested.

The recovered learning rates are (while still rather high) consistently lower than the ones fitted to behaviour. The differences in inverse temperature are even more striking, with recovered temperature being up to 10 x as high as the fitted values. I also think rather than presenting mean values, it would be more helpful to provide scatterplots and correlation coefficients in the supplements. I would like to emphasize that I think it is unlikely this would change the results a lot, as even somewhat deviant learning rates will likely lead to rather similar estimates for PE and $|PE|$, but it would still be helpful to have this information (scatter diagrams and correlation coeffs) presented. One last suggestion, for the new computational models, the model parameters are only described, maybe you want to consider presenting the equations for ease of understanding?

We have now implemented the changes suggested. We would like to apologize about the fact that there was a mistake in the original parameter recovery code (the nature of the block had not been considered). We have now fixed this error and have recreated the table.

Parameter estimate for best behavioural model, depicted as mean \pm SD and Recovery parameters estimates (across subjects) \pm SD. Parameters are presented below the ones found during the fitting procedure.

	HC	HA	VP	NP
Learning rate	0.7955 \pm 0.1445	0.8274 \pm 0.1961	0.6912 \pm 0.1648	0.6036 \pm 0.1206
	0.6658 \pm 0.1323	0.8241 \pm 0.0898	0.5878 \pm 0.0867	0.4642 \pm 0.0536
SoftMax inv. Temp	0.1992 \pm 0.2202	3.9277 \pm 2.2033	0.1190 \pm 0.2169	0.0388 \pm 0.1050
	0.0323 \pm 0.0578	1.0571 \pm 0.6581	0.0384 \pm 0.1132	0.0003 \pm 0.0000
Choice stickiness	-0.2351 \pm 0.2945	0.7982 \pm 0.2808	0.1908 \pm 0.3010	-0.0546 \pm 0.3120
	-0.3481 \pm 0.1474	-1.0000 \pm 0.0003	-0.2195 \pm 0.0950	-0.1155 \pm 0.1496

Supplementary table 1. Recovery parameters estimates (across subjects) \pm SD. Parameters are presented below the ones found during the fitting procedure.

The parameters are much more similar, which is to be expected. We have now added a figure as suggested but instead of scatter plots, have decided to separate the parameters according to the block (supplementary figure 1e [top row: fitted parameter; bottom row: recovered parameter]). The coefficient of correlation across subject was 0.45. We have also describe the method in the main text.

Supplementary Figure 1. (a) Summed log-model evidence using a variational inversion scheme for all models. Higher scores indicate better fit. (b) Nested model comparison using the summed integrated Bayesian Information Criterion (BICint) scores for all models. Lower scores indicate better fit. RL_HEP_SimC is the winning model. (c) Comparisons of

Bayes factors between best and second-best models. Bars are individual blocks sessions and direction of the bars (left vs right) indicate better model fit in favour of the RL_HEP_SimC (going to the right) compared to RL_HEP_rw (going to the left). (Small panel) Exceedance probability also favours the SimC model among our set of candidate models. The dashed red line indicates an exceedance probability of 0.95. (d) Model-predicted choice probabilities (y-axis) derived from the SimC algorithm (binned into four bins – bin size of 0.25 - and averaged across all subjects and across symbols) closely matched participants observed behavioural choices (x-axis), calculated for each bin as the fraction of trials in which they chose one colour. (e) Parameters fitted (top row) and recovered (bottom row) following parameter recovery procedure. The box plots present the mean, the SEM and 95% confidence intervals for the mean. Jittered raw data are plotted for each group.